# Phytohormone profiling in an evolutionary framework

Vojtěch Schmidt [1,2,4], Roman Skokan [1,4] ✉, Thomas Depaepe [3], Katarina Kurtović [2], Samuel Haluška [1,2], Stanislav Vosolsobě [2], Roberta Vaculíková[1], Anthony Pil[3], Petre Ivanov Dobrev[1], Václav Motyka[1], Dominique Van Der Straeten [3] & Jan Petrášek [1,2] ✉

The genomes of charophyte green algae, close relatives of land plants, typically do not show signs of developmental regulation by phytohormones. However, scattered reports of endogenous phytohormone production in these organisms exist. We performed a comprehensive analysis of multiple phytohormones in Viridiplantae, focusing mainly on charophytes. We show that auxin, salicylic acid, ethylene and tRNA-derived cytokinins including *cis*-zeatin are found ubiquitously in Viridiplantae. By contrast, land plants but not green algae contain the *trans*-zeatin type cytokinins as well as auxin and cytokinin conjugates. Charophytes occasionally produce jasmonates and abscisic acid, whereas the latter is detected consistently in land plants. Several phytohormones are excreted into the culture medium, including auxin by charophytes and cytokinins and salicylic acid by Viridiplantae in general. We note that the conservation of phytohormone biosynthesis and signaling pathways known from angiosperms does not match the capacity for phytohormone biosynthesis in Viridiplantae. Our phylogenetically guided analysis of established algal cultures provides an important insight into phytohormone biosynthesis and metabolism across Streptophyta.

Land plants evolved from within a group of freshwater green algae called charophytes. Together they form the lineage Streptophyta, which is embedded in Viridiplantae alongside the earlier-diverging chlorophyte and prasinodermatophyte algae[1]. The extant charophyte lineages are highly morphologically divergent, ranging from unicellular and filamentous to multicellular with differentiated cell types[2].

Phytohormones comprise several classes of endogenous, organic substances that influence physiological processes at lower concentrations than vitamins or nutrients[3]. These phytohormone classes include auxins, cytokinins (CK), jasmonates, ethylene, abscisic acid (ABA), salicylic acid (SA), strigolactones (SL), gibberellins (GA) and brassinosteroids (BR). Genomic evidence suggests that the machinery underlying ethylene and CK signaling evolved prior to the emergence of land plants[4], but it is currently unknown whether ethylene or CK serves as conserved signals between green algae and land plants. The canonical perception mechanisms of auxin, ABA, jasmonates, and SA were imposed on pre-existing transcriptional networks in the last common ancestor of land plants[4–6]. Finally, the receptors of SL, GA, and BR emerged at different points in the evolution of vascular plants[7–9]. While the gene families underlying CK biosynthesis predate plant terrestrialization, these are not easily distinguishable in green algae and bryophytes from the ancient mechanisms of tRNA modification and purine metabolism[10–12]. The canonical biosynthetic pathways (as known from angiosperms) of the other above-listed phytohormones were assembled in the last common ancestor of land plants or later in the lineage, even though

[1]Institute of Experimental Botany of the Czech Academy of Sciences, Rozvojová 263, 165 02 Prague 6, Czechia. [2]Department of Experimental Plant Biology, Charles University, Viničná 5, 128 44 Prague 2, Czechia. [3]Laboratory of Functional Plant Biology, Ghent University, K.L. Ledeganckstraat 35, B-9000 Ghent, Belgium. [4]These authors contributed equally: Vojtěch Schmidt, Roman Skokan. ✉e-mail: skokan@ueb.cas.cz; petrasek@ueb.cas.cz

homologs of certain genes are present in both charophytes and chlorophytes[5,6,13–15].

Genomic inquiries into the origins of phytohormone biosynthesis have been paralleled by profiling of endogenous phytohormones. First, it should be noted that the biosynthesis of many compounds recognized as phytohormones is not restricted to Viridiplantae[16–18]. In green algae, research to date has prioritized unicellular chlorophytes. Despite their distant relationship with land plants, chlorophytes were reported to produce nearly all the compounds known as bioactive phytohormones[19]. Charophytes, the closer living relatives of land plants, have attracted considerably less attention in phytohormone profiling. Results obtained from a few charophyte genera collectively attest that they can likewise produce compounds known as phytohormones[15,18,20–29]. However, phytohormone profiling in green algae and bryophytes is not without controversy, as individual studies do not always agree on the detection of specific compounds in equivalent biological material. Examples include abscisic acid in the chlorophyte *Draparnaldia* spp.[22,30], and jasmonic acid in the charophyte *Chara* spp.[26,28] and the bryophyte *Marchantia polymorpha*[20,21,31,32]. Three studies in the charophyte *Klebsormidium* spp. differed in both detection and endogenous concentrations of CK but closely agreed on the auxin indole-3-acetic acid (IAA)[18,25,27]. Individual studies tend to be limited in taxon sampling or the selection of analyzed compounds. Moreover, multiple lineages of charophytes have never been explored by phytohormone profiling, despite their important phylogenetic position in plant evolutionary research[33,34]. A systematic, analytical investigation of phytohormone biosynthesis and metabolism in Streptophyta that predate transition to land has never been performed.

To address the inconsistencies and gaps in analytical evidence, we provide a broad screen of compounds corresponding to multiple classes of phytohormones in a wide range of Viridiplantae. The selected taxa represent all families of charophyte algae, but also include several chlorophyte algae and land plants as outgroups. All investigated organisms were cultured under closely comparable conditions and subjected to the same analytical methods. We found that compounds known as auxin, CK, ABA, jasmonates, SA, and ethylene are produced in both charophytes and land plants, with some notable distinctions between the two lineages. In addition, several phytohormones were found to be released into the culture media. We discuss our data in the context of current hormonomics and genomics.

## Results

### Analytical design in an evolutionary framework

The algal and land plant strains were selected based on their phylogenetic position (Fig. 1), establishment in research as model organisms, availability in axenic cultures, and ease of cultivation (see Supplementary Data 1). Both biomass and the corresponding culture media have been sampled at a proliferative stage of growth, but multiple selected algae were likewise sampled at the stationary growth phase, particularly the charophytes. A liquid chromatography/mass spectrometry (LC/MS)-based method designed to detect the broadest possible spectrum of phytohormone compounds was applied to analyze the samples (full list of analytes in Supplementary Data 2). Additionally, the extracellular release of the gaseous phytohormone ethylene by living algal and plant biomass was investigated using laser-based photoacoustic spectroscopy. The results obtained were related to the same analysis performed on blank media containing no biological material.

### Phytohormones in biomass

The profiles of phytohormones and their metabolites measured in algal and land plant samples are summarized in Fig. 1 (see Supplementary Figs. 1–3 for more detailed versions). The raw dataset is provided in Supplementary Data 3 for all compounds except ethylene, which is detailed in Supplementary Data 4.

ABA was detected rarely in green algae, at low concentrations (pM to units of nM), and universally in the charophyte samples that were investigated at the stationary growth stage. By contrast, ABA was detected consistently in the thalli of the investigated land plants (Fig. 1, Supplementary Fig. 2). The concentrations in the lycophyte *Selaginella uncinata* were similar (order of $10^1$ nM) while those in bryophytes were lower (units of nM) compared to those previously measured in these lineages (Supplementary Data 5). Only the lycophyte contained the ABA catabolites phaseic and dihydrophaseic acids (PA & DPA). These results summarily show that ABA can be produced in charophytes at low concentrations in stationary-phase cultures, whereas it can be regularly detected in the growing cultures of land plants. Moreover, only the tested lycophyte showed evidence of active ABA metabolism under our conditions.

The auxin profile in our dataset was characterized by the omnipresence of free IAA, its catabolite 2-oxo-IAA (oxIAA), and the frequent presence of the IAA precursor indole-3-acetamide (IAM). Land plants differed from green algae by additionally containing the amide- and ester-conjugates of IAA (Fig. 1), namely IAA-glutamate (IAA-Glu) and IAA-glucosyl ester (IAA-GE). The endogenous concentrations of IAA ranged toward the lower end compared to the available literature, where these span the entire nanomolar scale (Supplementary Data 5). The phenolic compound phenylacetic acid (PAA) is known to have a weak auxin activity and is widespread in the tree of life[35]; it was accordingly omnipresent in our dataset. To conclude, we found that free IAA, PAA, and some compounds related to both were present in all the organisms investigated. However, only land plants showed evidence of active IAA metabolism by conjugation into amides and glucosides.

Among CK, we found that both chlorophyte and charophyte green algae contained only the tRNA-related CK types, namely $N^6$-($\Delta^2$-isopentenyl)-adenine (iP), *cis*-zeatin (cZ) and methylthio CK (MeS-zeatin and MeS-iP). Land plants likewise contained these, but additionally produced the *trans*-zeatin (tZ) type CK and cZ/tZ-O-glucosides. The last CK type analyzed, dihydrozeatin (DZ), as well as CK-N-glucosides were absent in all strains analyzed (Fig. 1). The measured CK concentrations varied by strain but were mostly similar between green algae and land plants and within the ranges reported previously (Supplementary Data 5), i.e., from pM to units of nM. In summary, we found that all the tested Viridiplantae contained CK derived from the post-transcriptional modifications of tRNA. However, only land plants produced the tZ-type CK and CK-O-glucosides, the latter indicating active CK metabolism.

The analysis of oxylipins/jasmonates (henceforth 'jasmonates') in the selected Viridiplantae revealed a particularly patchy profile. A few individual strains produced copious amounts, whereas other strains (including all four tested chlorophytes) contained none of the tested jasmonates (Fig. 1). The most frequently detected and simultaneously the most abundantly produced jasmonate was dinor-12-oxo-phytodienoic acid (dnOPDA), with two strains of *Coleochaete scutata* containing particularly high (μM) endogenous concentrations (Supplementary Data 3). dnOPDA was altogether found in four charophytes, while its detection in *Marchantia polymorpha* represented the only case of jasmonate occurrence in our tested land plants. The charophytes *Coleochaete scutata* and the two strains of Klebsormidiophyceae constituted the only organisms that produced jasmonates other than dnOPDA, namely OPDA and jasmonic acid (JA). In Klebsormidiophyceae, JA was only found in the stationary-stage cultures. Jasmonoyl-isoleucine (JA-Ile), the bioactive jasmonate of euphyllophyte vascular plants[15], was absent in our entire dataset. The measured concentrations ranged from nM (JA, OPDA) to μM (dnOPDA in some strains). The levels of dnOPDA in *Marchantia polymorpha* and JA & dnOPDA in *Klebsormidium nitens* in our analysis were similar to those

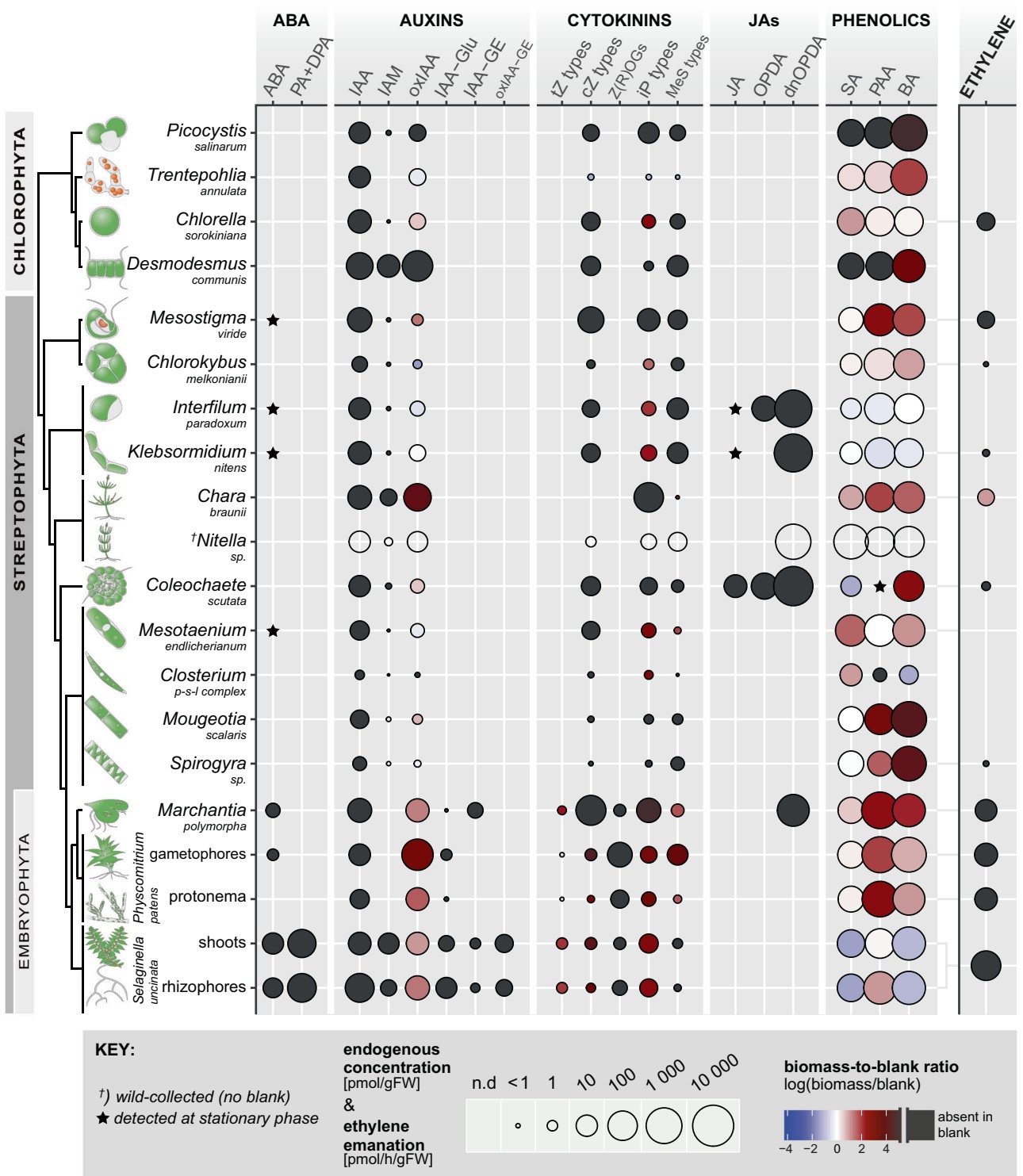

**Fig. 1 | Endogenous phytohormone compounds detected in the biomass of green algae and land plants and ethylene emanation.** Circle size denotes mean concentration in biomass (pmol per gram fresh weight) and mean ethylene emanation (rightmost column; pmol/h/gFW). No circle, compound not detected (n.d.). Color code denotes the ratio between the concentrations measured in biomass and blank medium (the latter containing no biological material), expressed in logarithmic scale. Blue shading, compound(s) prevalent in blank. Red shading, compound(s) prevalent in biomass. Black, compound(s) absent in blank. Symbols: star, compound only detected in stationary-phase cultures; cross, wild-collected biological material (no blank available). ABA abscisic acid, PA phaseic acid, DPA dihydrophaseic acid, IAA indole-3-acetic acid, IAM indole-3-acetamide, oxIAA 2-oxo-IAA, IAA-Glu IAA-glutamate, IAA-GE IAA-glucose ester, tZ *trans*-zeatin, cZ *cis*-zeatin, Z(R)OGs zeatin (riboside)-*O*-glucosides; both *cis*- and *trans*- isomers, iP *N*[6]-(Δ[2]-isopentenyl)-adenine, MeS methylthio, JA jasmonic acid, OPDA 12-oxo-phytodienoic acid, dnOPDA dinor-OPDA, SA salicylic acid, PAA phenylacetic acid, BA benzoic acid. Minimum *n* = 3 for biomass (independent cultures; exact sample size for each strain is listed in Supplementary Data 3), *n* = 2 for blank media. Ethylene measurements for both biomass and blank media were performed in *n* = 5. Data variation is shown in Supplementary Figs. 3 and 4 and listed in Supplementary Data 8.

reported[15], although our two studies differed in OPDA detection. In conclusion, we observed several cases of jasmonate production in the tested streptophytes, plausibly indicative of a conserved reaction to the culture conditions but with lineage-specific sensitivity. dnOPDA was detected in five of the investigated strains, while only three of these strains additionally contained ODPA and/or JA.

SA was detected in all tested species, with no apparent differences between land plants and green algae (Fig. 1). We likewise observed the omnipresence of benzoic acid (BA), which can serve as an immediate SA precursor[5], but also is a more general primary metabolite[36]. The measured SA concentrations ranged on the nanomolar scale, as previously reported in green algae, bryophytes, and many vascular plants (Supplementary Data 5)[5]. In short, we observed that all tested Viridiplantae produced SA and BA.

As ethylene is a gaseous, diffusable phytohormone, we measured its accumulation in enclosed vials containing biomass and culture medium over a period of 6 hours using laser-based photoacoustic spectroscopy. A reduced sample set was analyzed compared to the LC/MS screen. Both land plants and charophytes released ethylene into the environment (Fig. 1 and Supplementary Fig. 4), although the latter at much lower amounts (1–2 orders of magnitude). The exceptions among algae are, peculiarly, the chlorophyte *Chlorella sorokiniana* and the early-diverging charophyte *Mesostigma viride*, both of which released ethylene at levels similar to land plants. The rate of ethylene release by *Spirogyra sp.* was closely comparable to an analogous measurement performed previously in *Spirogyra pratensis*[29]. Hence, we found that representatives across Viridiplantae can produce ethylene. However, land plants produced much higher amounts compared to the more closely related charophyte lineages.

Last but not least, we note that our LC/MS method allows for the detection of certain GA, BR, and SL (Supplementary Data 2), although these compounds are typically measured using optimized purification, extraction, and analysis procedures. We detected none of the analyzed compounds under our approach, although other studies had previously reported their detection in similar biological material. This includes GA in unicellular chlorophytes, *Chara* spp. and *Selaginella moellendorffii*[37–39], BR in unicellular chlorophytes[40] and SL in bryophytes[24,41].

## Phytohormones in culture and control media

Phytohormones are recognized for developmental control in the three-dimensional bodies of land plants with differentiated tissues and cell types. Most green algae do not possess this morphological complexity, leading to discussions about whether an ancestral function of phytohormones, if present in ancient or extant green algae, might be facilitated via their release into the environment[42,43]. This motivated us to analyze the content of phytohormone compounds not only in biomass, but also in the corresponding culture media. The results are summarized in Fig. 2a (raw data in Supplementary Data 3) and henceforth described.

CK and phenolics (SA, PAA, BA) were detected in the culture media from the majority of investigated strains. CK levels in culture media were lower compared to biomass. Phenolics varied in their prevalence between culture media or biomass by the specific analyte and individual organism. Auxins were detected less frequently in the culture media compared to biomass, and the ratio between the two again varied by analyte and species. While free IAA was detected in the culture medium of half the investigated organisms, oxIAA was nearly omnipresent. Among green algae, ABA was only found in the culture medium of charophytes that also contained it intracellularly. By contrast, land plants contained endogenous ABA but did not release it into the culture media. The only cases of jasmonate content in the culture media were represented by JA in the charophytes *Interfilum paradoxum* and *Coleochaete scutata*, which also contained it intracellularly.

Further context to the measurements performed in biomass and culture media is gained from the same analysis in blank media, which contained no biological material. We revealed that IAA, ABA, jasmonates, and most CK were absent in the blank media (black circles in Figs. 1 and 2b). Hence, their presence in either biomass or medium resulted from the activity of living organisms. Ethylene release was likewise detected only in the setups containing living material except for one positive case in the blank medium of *Chara braunii* (Supplementary Data 4), which might be explained as an artifact of the medium composition. Certain other analytes were frequently detected in the blank media. This includes oxIAA and iP-type CK, which were present at lower levels in blanks compared to biomass (Fig. 1) and the culture media (Fig. 2b), suggesting that the living organisms additionally produced and released these compounds on their own. Phenolics were likewise present in blanks, and the ratio toward the content in biomass and culture media varied by analyte and species. For instance, *Mougeotia scalaris* and *Mesotaenium endlicherianum* produced PAA and released it into the environment. Klebsormidiophyceae enriched the culture medium with PAA and BA but did not accumulate these intracellularly. *Selaginella uncinata* degraded SA and BA present in the medium upon inoculation, possibly by uptake and internal metabolization. Summarily, we found that the detection of ABA, auxins, CK, jasmonates and ethylene in either biomass or culture medium could be entirely or predominantly attributed to the action of living organisms. Phenolics were not only produced by the living biomass but frequently also had been a part of the culture media prior to inoculation with living cultures, and individual species differed in their production, uptake, and release.

The content of certain phytohormones in blank media motivated us to further address the possible sources of these contaminants by analyzing different purity grades of water and three brands of agar (Fig. 2c). Among all analyzed compounds, the water samples contained only the phenolic BA. However, BA was the only analyte in our analysis which lacked the qualifier ion, so the signal might not be entirely specific to BA. The agar samples (incorporated in MilliQ grade water) contained oxIAA, iP-type CK, and considerable amounts of the phenolics BA, SA, and PAA (Fig. 2c, Supplementary Data 3). Hence, the low-level detection of additional compounds in blank media compared to these agar and water controls is likely to be attributed to different, unknown sources of contamination.

## Phytohormones and biological contamination

The issue of contamination of algal material by eukaryotic or prokaryotic microorganisms in phytohormone measurements and treatments has long been debated[35,44]. In our effort to obtain axenic cultures whenever possible (see Supplementary Fig. 5 for the culture contamination test), we managed to obtain three sets of charophyte strains, corresponding to three charophyte lineages. Each set contained two strains from the same lineage, one axenic and the other contaminated. These sets constituted two strains of *Mesostigma viride* (Mesostigmatophyceae), *Chlorokybus cerffii* and *C. melkonianii* (both Chlorokybophyceae), and two strains of *Coleochaete scutata* (both Coleochaetophyceae). Remarkably, we found that phytohormone profiles in the biomass were qualitatively similar within lineages, regardless of strain contamination (Supplementary Fig. 6). Some qualitative differences could be observed in the culture media, but these were lineage-specific (Supplementary Fig. 6). For instance, the contaminated *Chlorokybus cerffii* culture medium contained IAA, in contrast to the axenic strain *Chlorokybus melkonianii*. However, the opposite pattern could be observed between the axenic and contaminated strains of *Mesostigma viride* and *Coleochaete scutata*. This could be attributed to a lineage-specific reaction to contamination among charophytes, or the activity of different (though unidentified) microorganisms present in these cultures. In summary, we did not detect any shared patterns in phytohormone biosynthesis and

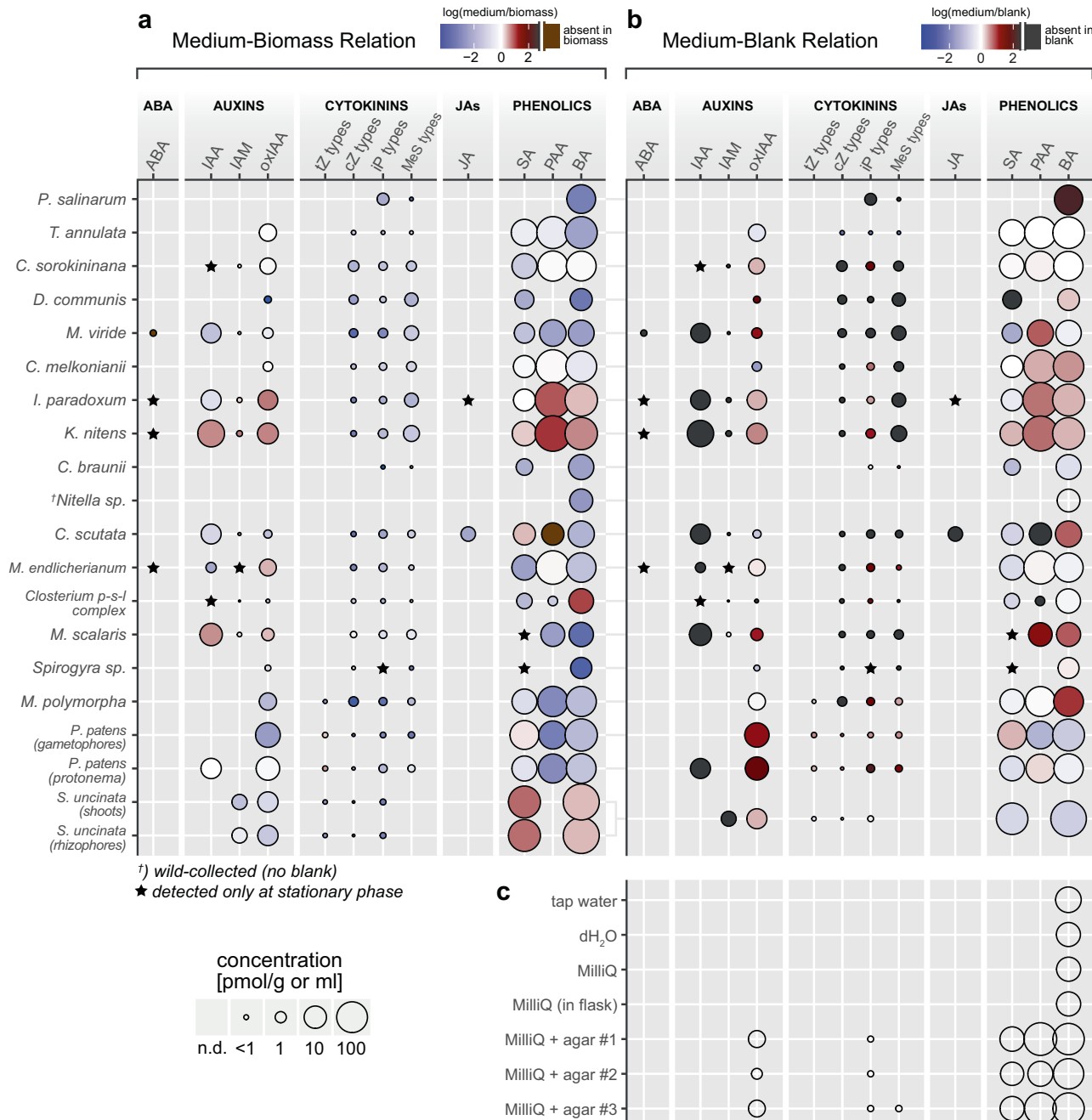

**Fig. 2 | Detection of phytohormone compounds in culture media and control samples of water and agar. a** Ratio between concentrations detected in culture medium and the corresponding biomass, in logarithmic scale. Color code: blue shading, compound prevalent in biomass; red shading, compound prevalent in medium; black, compound absent in biomass. **b** Ratio between concentrations detected in culture medium and blank medium (containing no biological material), in logarithmic scale. Color code: blue shading, compound prevalent in blank; red shading, compound prevalent in medium; black, compound absent in blank. The circle size in **a**, **b** denotes the mean concentration in culture media (pmol per gram or ml). No circle, compound not detected (n.d.). Minimum $n = 3$ for culture media (independent cultures; exact sample size for each strain is listed in Supplementary Data 3), $n = 2$ for blank media. **c** Compounds detected in different purity grades of water and three brands of agar (1.5% w/v, incorporated in MilliQ grade water). The circle size denotes mean concentration (pmol per gram or ml); $n = 3$ samples. No circle, compound not detected (n.d.). Compound abbreviations are listed in the legend in Fig. 1. Data variation is shown in Supplementary Fig. 3 and listed in Supplementary Data 8.

excretion in response to microbial contamination. This suggests that lineage-specific patterns are likely more profound than the influence of microbial contamination.

## Discussion

Comparing our results to the available literature on phytohormone profiling, we see that certain compounds are being detected more consistently than others. Our findings align with the ubiquitous occurrence of IAA and oxIAA throughout Viridiplantae (Supplementary Data 5), much like the phenolic phytohormone SA[5]. IAA-amides were absent in green algae in our screen and certain other studies[25,26], though isolated occurrences at low amounts had been reported elsewhere[18]. The detection of the jasmonate JA had previously varied in charophytes and bryophytes, while OPDA and dnOPDA were found to

be more consistent (Supplementary Data 5). We detected endogenous jasmonates in multiple charophytes, often at considerable concentrations, but found land plants to be nearly devoid of jasmonates under similar culture conditions. As suggested by the role of ODPA and dnOPDA in a thermotolerance response conserved in Streptophyta[15,45], the jasmonate-rich charophyte strains in our analysis may have been growing outside their temperature optimum. Additional stimuli such as wounding regulate jasmonate production in land plants[15], plausibly accounting for their near-absence in land plant material under our conditions. The published CK profiles in non-seed Viridiplantae have been inconsistent at the lineage level. Despite numerous qualitative disagreements (see Supplementary Data 5), our results at least support two general trends in green algae and bryophytes: CK profiles are dominated by the tRNA CK types, and CK-*N*-glucosides are absent[18,19,46]. We detected ABA in all the tested land plants but only a few green algae, whereas ABA presence in green algae and non-Viridiplantae lineages has been well-established in literature[17,26,27,42,47–49]. Since ABA was only found in stationary-phase chyrophyte samples, we speculate that ABA production at detectable levels in green algae may result from certain stress conditions that were rarely achieved among our cultured strains. In summary, we confirm that free auxins, tRNA CK, and SA are produced universally in green algae, bryophytes, and lycophytes. By contrast, the biosynthesis of ABA in charophytes and jasmonates in Streptophyta at detectable levels appears to depend on certain stimuli, in our case manifested from culture conditions. We found *t*Z and IAA- and CK-conjugates to be restricted to land plants, but conflicting reports had been published. Regarding ethylene release, to our knowledge, current literature lacks data comparable to our dataset.

Our LC/MS profile in green algal and land plant biomass was complemented with the same analysis performed in the corresponding culture media. CK profiles in biomass were generally mirrored at lower concentrations in culture media, supporting earlier isolated reports of extracellular CK release in Viridiplantae[50]. It has been reported that chlorophytes excrete ABA and IAA[49,51–54], which is contrary to our findings. All chlorophytes in this study also lacked intracellular ABA. On the other hand, charophytes that contained intracellular ABA were also excreting it into the media. The excretion of IAA in charophytes was previously proposed to reflect an ancestral function in inter-organismal communication[43,55]. We measured increased IAA concentrations in the culture medium of *Klebsormidium nitens* compared to biomass, coinciding with the conserved auxin efflux function of the PIN-FORMED (PIN) proteins in the genus[56]. However, we also found that cases of auxin excretion among charophytes did not universally correlate with PIN conservation[43]. In the control measurements of agar samples, the content of auxins, CK, and SA may reflect the original biological source in agarophyte red algae[57–59]. Although culturing plants on agar is equivalent to treatment with ca. 10–20 nM SA, the effective doses used in treatments of plants are much higher[60]. In all, we revealed that phytohormones were released into the culture media in streptophytes. This mainly included auxin, CK, and phenolics, with lineage-specific differences. Additionally, we show that certain phytohormone compounds are present at basal levels in culture media as contaminants from the media components.

Our comparison of certain axenic vs. xenic (contaminated) charophyte cultures revealed little influence of contaminants on the overall phytohormone profiles. No conserved reaction to contamination was revealed among three investigated charophyte lineages, in either intracellular or extracellular phytohormone content. As for the contaminating microorganisms, little interpretation could be made; previous studies on phytohormone biosynthesis and responses in bacteria and fungi have been focused dominantly on the strains associated with vascular plants[61–63]. Although our results do not

suggest that the recognized phytohormones are relevant in charophyte-microorganism interactions, a dedicated future investigation should include the isolation and identification of microbial strains living in natural association with the algae. Natural multi-algal consortia could be studied in a similar fashion.

Plant evolution is researched in large part by tracking the conservation of genes known from angiosperms. However, phytohormone profiles in Viridiplantae regularly do not match genomic inferences. IAA and oxIAA are omnipresent in Viridiplantae, whereas we only detected glucosylated IAA in bryophytes and a lycophyte. Yet, the enzymes recognized for auxin biosynthesis, non-decarboxylative IAA oxidation, and IAA glucosylation, respectively, are only conserved in land plants, angiosperms, and seed plants[13]. Inversely, the charophyte *Klebsormidium* spp. lacks endogenous IAA amides (this study and ref. [18]) despite uniquely containing gene homologs of IAA-amide synthases, which are otherwise restricted to land plants[13]. CK are well known to regulate development in bryophytes via signaling conserved with angiosperms[64], but no such knowledge exists in charophytes. Correspondingly, we found only land plants among the sampled organisms to contain *t*Z-type CKs, which are known to be highly bioactive[65]. However, these investigated land plants lack the adenylate-type isopentenyltransferases recognized for *t*Z biosynthesis, i.e., do not differ from green algae in the genomic evidence for CK biosynthesis[12]. To make matters more complicated, green algae can process applied IAA into IAA-amides and break down applied *t*Z into adenine and adenosine, despite lacking IAA-amidases and CK dehydrogenases[18,23]. We found all but one of the investigated Viridiplantae to release ethylene into the environment. Yet, the enzyme directly responsible for ethylene biosynthesis is only conserved in seed plants[14]. Two SA biosynthesis pathways are deeply conserved in land plants. Among green algae, only certain charophyte lineages contain all the known components of at least one (the β-oxidation-based) of these pathways[5]. Regardless, SA is produced ubiquitously in Viridiplantae (our study and[5]). Interestingly, the chlorophyte *Chlamydomonas reinhardtii* lacks some genes of the β-oxidation-based SA biosynthesis pathway, but a mutation in one enzyme conserved with the angiosperm *Oryza sativa* results in decreased endogenous SA levels in both species[5]. These examples remind us there is much left unknown in the lineages outside of seed plants, including secondary metabolism[6]. More research into these organisms will help us overcome the limits of viewing plant evolution through the prism of knowledge gained mainly in angiosperms.

The question of the origins of phytohormone function has been approached by analytical profiling of native compounds, treatments by externally applied phytohormones, and comparative genomics. Biotechnology-motivated studies have revealed that unicellular chlorophytes natively produce the most known phytohormones and show growth responses to their external application. Similar results were obtained in "microalgae" from non-Viridiplantae lineages, including Eustigmatophyceae, Euglenozoa, or Cyanobacteria (reviewed in refs. [19], [66], [67]). As phytohormones are defined as "natural compounds affecting physiological processes at low concentrations"[3], these observations would suggest that phytohormone identity of multiple organic compounds is widespread in the tree of life. However, no underlying mechanism, genetic or otherwise, conserved between land plants and any of these lineages has yet been shown. Research through comparative genomics provides a different perspective; the molecular machineries underlying the responses to most phytohormones only became operational in the last common ancestor of land plants or later in this lineage[4–9]. Our results showed that land plants but not green algae (chlorophytes or charophytes) consistently contained ABA and produced auxin and CK metabolites. Other studies focusing on these hormones have likewise found differences between land plants and green algae. In treatments, lower micromolar doses of

ABA, auxin, and CK elicit growth phenotypes in land plants such as bryophytes[68–72], but comparably higher doses of auxin and ABA (and much higher than the levels detected as endogenous in this study) are necessary to achieve comparably milder effects in charophytes[73–76]. No data are available regarding CK treatments in charophytes. Comparative genomics revealed that the genes coding for auxin and ABA receptors are functionally conserved across land plants but are absent in charophytes[71,72,77,78], whereas the CK receptor homologs present in charophytes are related to a clade that does not act in CK perception in land plants[12]. Additionally, gene homologs of CK signaling in the charophyte *Spirogyra pratensis* are not transcriptionally responsive to applied CK[79]. Collectively, our results together with the above-listed observations imply that ABA, auxin, and CK might only have assumed phytohormone identity in an ancestral land plant. It must be noted that much more work lies ahead of us to properly understand the evolution of phytohormone responses. Recent research has revealed that both charophytes and land plants utilize OPDA and dnOPDA in a thermotolerance response, which is distinct from the role of these compounds in jasmonate signaling that only evolved in land plants[15,45]. Canonical auxin signaling is likewise a land plant invention[77], yet applied IAA elicits rapid changes in plasma membrane potential and protein phosphorylation in both land plants and charophytes; while the native function in charophytes remains unknown, it may involve a reaction to light or osmotic stress[80]. As charophytes become better explored and established as model organisms, we are likely to be updating our views regarding the evolution and origins of phytohormone responses.

How can we interpret the presence of phytohormones in organisms that lack a bona fide phytohormone response? The similar profiles of IAA and tRNA CK in many green, red, and brown algae, fungi, animals, and other organisms[17,18,48,57,81–83] might simply reflect the metabolism of indoles and purines/tRNA[84,85]. Similarly, the presence of ABA in algae such as *Dunaliella* spp. may be a side-effect of carotenoid metabolism[22,86,87]. Increased ABA production from carotenoid oxidation under stress conditions has been proposed as an evolutionary prerequisite to ABA functioning as a stress hormone[6]. Indeed, extant green algae can accumulate ABA under stress without signs of ABA itself affecting stress responses[88]. On a related note, it may be worth considering that the compounds recognized as phytohormones, such as IAA or CK, might only reflect phytohormone action when natively present in amounts that exceed specific thresholds. After all, phytohormone function in angiosperms follows a strict dose-response relationship and is often coupled with their considerable accumulation in specific tissues and developmental stages[89–92]. Therefore, if compounds known as phytohormones have physiological functions in extant charophytes, conserved with land plants or not, these may be spatiotemporally or developmentally restricted (e.g., to developing zygospores). Uncovering such possible functions would yet require a considerable amount of research.

In the past two decades, bryophytes have been the torch illuminating the dark corners of plant evolutionary research. A new frontier is currently opening in charophyte algae, perhaps even more challenging due to longer divergence times coupled with considerable diversity in morphology and physiology. We revealed that charophytes are capable of producing most organic compounds known as active phytohormones in land plants. However, certain patterns in biosynthesis and metabolism coupled with genomic and empirical evidence indicate that major, perhaps fundamental innovations underlying phytohormone action have only evolved after the transition to land. Hence, this study represents one piece in the puzzle of the origins and evolution of phytohormone responses.

## Methods
### Chemicals
Supplementary data 6 lists the chemicals used in this study.

## Algal and plant strains and cultivation
Algal strains were obtained from the following institutions: Microbial culture collection, National Institute for Environmental Studies, Tsukuba, Japan (NIES); Culture collection of algae, Goettingen University, Germany (SAG); Central collection of algal cultures, Duisburg University, Essen, Germany (CCAC); Culture collection of algae, University of Texas at Austin, USA (UTEX); Culture collection of algae, Department of Botany, Charles University, Prague, Czechia (CAUP). *Selaginella uncinata* ("Comenius", original strain) was obtained from the Botanical garden at the Comenius University, Bratislava, Slovakia, and established as a sterile culture. The wild-collected samples of *Nitella* sp. and the surrounding water (flash-frozen in liquid $N_2$ in situ) were obtained from a freshwater spring pond under sandstone rocks (GPS: 50.6402767 N, 14.5147078E) in April 2021; the surrounding water was notably pure and the algae not obviously covered with epiphytic microflora.

See Supplementary Data 1 for a complete list of strains with source identifiers and the culture media used. Murashige–Skoog medium (Duchefa M0221), Gamborg medium (Duchefa G0210), and Bold's Basal Medium (BBM; Merck B5285) were purchased commercially. BCD medium supplemented with diammonium tartrate (BCDAT) was prepared according to[93]. C medium, Pro medium, and a modified version of the SWCN-4 medium for Charophyceae were prepared per instructions from the NIES collection (https://mcc.nies.go.jp/02medium-e.html). Modified SWCN-4 as follows: garden soil was mixed with river sand (1:4), dampened with $dH_2O$, autoclaved 3×, laid into sterile test tubes (3 cm height), and supplemented with 40 ml sterile $dH_2O$. BBM, C, and Pro media were supplemented with customized vitamin doses: $B_{12}$ 10 mg/l medium, $B_7$ 2 mg/l, $B_1$ 10 mg/l. Individual vitamin stocks (1000×) were dissolved in $dH_2O$, sterilized by filtration (Millex SLGS033SS), stored at −20 °C, and added into cooled-down but not yet solid agar-supplemented medium, or upon algal inoculation (liquid medium). All media were sugar-free.

Land plants were cultured on solid media in plates sealed with surgical tape (Micropore 1530-0). *Marchantia polymorpha* were inoculated from individual gemmae. *Physcomitrium patens* protonema were homogenized weekly using Ika T25 Digital Ultra Turrax with 8 G disperser tool (IKA-Werke, Germany) and spread on plates containing medium overlaid with cellophane foil. When necessary, cultures were left to grow without homogenization to allow gametophore emergence. *Selaginella uncincata* (chosen for its superb growth in vitro) was inoculated from apical branch cuttings. 2–3 apical explants of *Chara braunii* (each with at least two nodes) were inoculated per one test tube containing soil and culture medium. Other algae, if not grown for analysis, were inoculated into the fresh medium from a fraction of the original culture (2–10% biomass) in 6–8 week intervals or longer, depending on the growth character of each strain.

Although we have generally avoided this measure in cultures intended for phytohormone analysis, certain algal strains (Supplementary Data 1) require supplementation of the culture media with soil extract for continuous proper growth. Soil extract was prepared by mixing soil with $dH_2O$ (1:3 volume), autoclaving, passing through filter paper overnight, and autoclaving again. The soil (without obvious leaf litter) was collected in a submontane forest of European beech (*Fagus sylvatica*; GPS: 50.8653578 N, 15.1046144E) at the end of winter.

All living material was cultured at 23 °C, 16:8 hours light:dark regime. The cultures were illuminated by mixed-spectrum fluorescent light tubes using Osram Fluora TLD 36 W (Osram Licht AG, Germany) and Philips Master TLD Super 36 W (Koninklijke Philips N.V., The Netherlands); see Supplementary Data 7 for details on illumination spectra and intensity.

## Culture contamination tests
The axenic status of living cultures was examined by two approaches. First, the algal biomass used for analysis was observed under a

microscope (Supplementary Fig. 5). Second, the algal and land plant biomass was inoculated on four types of media designed to support the growth of bacteria and fungi. These media were labeled as LB, Euglena, Trebouxia, and Sabouraud (all supplemented with 1.5% w/v agar, Duchefa P1001) and were prepared per instructions of the Culture Collection of Algae at the University of Texas, Austin (UTEX) (https://utex.freshdesk.com/support/solutions/articles/19000006667-how-do-you-test-for-contamination-). After inoculation, the plates were sealed with parafilm and cultured in the dark at 37 °C for 7 days. All plates were photographed and examined microscopically (Supplementary Fig. 5).

## Sample preparation

Algal biomass was sampled during the proliferative phase of culture growth (1.5–3 weeks after inoculation, depending on individual strains). Streptophyte algae were additionally sampled during the stationary phase of growth (4–6 weeks). *Marchantia polymorpha* thalli were sampled on day 19. *Physcomitrium patens* protonema and gametophores were sampled after 1 and 5 weeks of growth, respectively. *Selaginella uncinata* was sampled after 5 weeks of growth.

Samples for LC/MS were harvested as follows (see Supplementrary Data 1): liquid-cultured algae were collected by pipetting (cutoff tips) and filtration through a nylon mesh filter (20 μm, Merck Millipore NY2004700) using underpressure. Strains with smaller cells, i.e., *Picocystis salinarum*, *Desmodesmus communis*, *Mesostigma viride*, and *Closterium p-s-l complex* were collected by centrifugation (1000 × *g*) and supernatant decantation. Algae and moss protonema growing on solid media were scraped with a spatula and filtered to remove excessive moisture. *Chara braunii* and *Nitella* sp. thalli (without rhizoids) were washed with dH$_2$O, and excessive moisture was removed with a cotton pad. Bryophyte thalli and *Selaginella uncinata* apical cuttings were directly transferred into collection tubes. Biomass samples (10-50 mg fresh weight each) were transferred into 2-ml thick-walled microcentrifuge tubes (SSIbio 2340-00, Scientific Specialities, USA) with screwable lids (SSIbio 2002-00), flash-frozen in liquid nitrogen and stored at −80 °C. Liquid culture media were sampled by pipetting, centrifugation (1000 × *g*), and decanting as 100 μl samples. Samples of solid culture media were scraped from inside the solid agar (avoiding any residual biomass) with a cut pipette tip and sampled as ca. 50 mg each. Blank media were sampled similarly. The media samples were likewise flash-frozen in liquid nitrogen and stored at −80 °C.

## Phytohormone analysis

Samples were extracted with 100 μl 1 M formic acid solution. The following isotope-labelled standards were added at 1 pmol per sample: $^{13}C_6$-IAA (Cambridge Isotope Laboratories, Tewksbury, MA, USA); $^2H_4$-SA (Sigma-Aldrich, St. Louis, MO, USA); $^2H_3$-PA, $^2H_3$-DPA (NRC-PBI); $^2H_6$-ABA, $^2H_5$-JA, $^2H_5$-tZ, $^2H_5$-tZR, $^2H_5$-tZRMP, $^2H_5$-tZ7G, $^2H_5$-tZ9G, $^2H_5$-tZOG, $^2H_5$-tZROG, $^{15}N_4$-cZ, $^2H_3$-DZ, $^2H_3$-DZR, $^2H_3$-DZ9G, $^2H_3$-DZRMP, $^2H_7$-DZOG, $^2H_6$-iP, $^2H_6$-iPR, $^2H_6$-iP7G, $^2H_6$-iP9G, $^2H_6$-iPRMP $^2H_2$-GA19, $(^2H_5)(^{15}N_1)$-IAA-Asp and $(^2H_5)(^{15}N_1)$-IAA-Glu (Olchemim, Olomouc, Czech Republic). The extracts were centrifuged at 30,000 × *g* at 4 °C. The supernatants were applied to SPE Oasis HLB 96-well column plates (10 mg/well; Waters, Milford, MA, USA) conditioned with 100 μL acetonitrile and 100 μl 1 M formic acid using Pressure+ 96 manifold (Biotage, Uppsala, Sweden). After washing the wells three times with 100 μl water, the samples were eluted with 100 μl 50% acetonitrile in water. The eluates were separated on Kinetex EVO C18 HPLC column (2.6 μm, 150 × 2.1 mm, Phenomenex, Torrance, CA, USA). Mobile phases were as follows: A, 5 mM ammonium acetate, and 2 μM medronic acid in water; B, 95:5 acetonitrile:water (v/v). The following gradient was applied: 5% B in 0 min, 5-7% B (0.1–5 min), 10-35% B (5.1–12 min), and 35-100% B (12-13 min), followed by a 1 min hold at 100% B (13-14 min) and return to 5% B. Hormone analysis was performed with an LC/MS system consisting of UHPLC 1290 Infinity II (Agilent, Santa Clara, CA, USA) coupled to 6495 Triple Quadrupole Mass Spectrometer (Agilent, Santa Clara, CA, USA). The Jet Stream (AJS) ion source parameters included: gas temperature 180 °C, gas flow 19 l/min, sheath gas temperature 400 °C, sheath gas flow 12 l/min, nebulizer pressure 25 psi, capillary voltage: positive/negative −3000 V/2500 V, nozzle voltage: positive/negative −0 V/1000 V. The compounds were analysed in multiple reaction monitoring modes (transitions are listed in Supplementary Data 2), with quantification by the isotope dilution method. Data acquisition and processing were performed with Mass Hunter software B.08 (Agilent, Santa Clara, CA, USA). Data processing was performed using Agile2 integration algorithm followed by manual verification. For calculation of concentrations reflecting the amount of sample and internal standards, see Supplementary Data 3.

For ethylene measurements, we adapted an established protocol[94]. Blank media were processed first to determine any possible background levels or noise and enable subsequent correction of the biomass samples. A set volume (Supplementary Data 4) of liquid or solid medium in clear-glass 10-ml chromatography vials (Thermo Fisher 10623633) was left standing (lid open) for 1 hour. The vials were then sealed air-tight with a rubber septum (Thermo Fisher 11845060) and a snap cap (Thermo Fisher 11805020) using a manual crimping tool (Thermo Fisher 10372525), then kept at 21 °C in low light conditions (35 μmol m$^{-2}$ s$^{-1}$) under 4 fluorescent tubes (Philips TLD 36 W/33-640). After 6 hours, ethylene levels within the headspace were measured using laser-based photoacoustic spectroscopy (ETD-300, Sensor Sense) in stop-and-flow mode. Algal biomass samples were suspended in fresh medium (same volume as used in blank measurements, see Supplementary Data 4) and transferred into the chromatography vials. Land plant samples were transferred into chromatography vials containing solid culture medium and a minimum volume of pure water (MilliQ grade) (Supplementary Data 4); no pure water was added to *Physcomitrium patens* gametophores, which were previously cultured in the vials. The biomass samples were then processed as the blank media, i.e., left open to acclimatize for 1 h, sealed for 6 h, and measured for total ethylene accumulated. Samples were measured in 5 biological and 5 blank medium replicates. The system was calibrated using standard ethylene mixtures (Air Liquide 23209012).

## Presentation of results and statistical evaluation

Phytohormone profiles are presented as means and related to other sample categories by logarithmic ratio to visualize possible background of compounds in media (relation to corresponding blank media), or possible excretion of compounds to media (relation to culture media). Relative errors of biological and technical replications of LC/MS analysis are provided in Supplementary Fig. 3 and Supplementary Data 8. The significance of cultivation effects (e.g., culture stage and strain axenicity; Supplementary Figs. 2 and 6) were evaluated by linear mixed effects model after logarithmic transformation (lme4 package[95]) and tested by Likelihood-ratio test ('lmerTest' package[96]). Differences between groups were determined by post-hoc comparison (Tukey method, 'emmeans' package[97]. The same statistical evaluation was used to determine the significance of ethylene emanation by the biomass, due to the presence of media during ethylene measurements on living material (Supplementary Fig. 4). Statistical comparisons of LC/MS data between biomass/culture media and blank media were not performed, as these were analyzed as separate samples. Statistical analyses and data plotting were performed using R software package 4.2.3 (R Core Team) with 'tidyverse' package environment (v2.0.0). Figures were finally composed using Adobe Illustrator 2020.

## Reporting summary

Further information on research design is available in the Nature Portfolio Reporting Summary linked to this article.

## Data availability
All raw LC/MS data generated in this study have been deposited in the Zenodo repository: https://doi.org/10.5281/zenodo.10411071. Source data are provided with this paper.

## Code availability
Code for data plotting and statistical analyses is deposited at the following links: https://github.com/VoSchmidt/BubblePlots-for-phytohormones https://github.com/vosolsob/phytohormone.

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

## Acknowledgements
We acknowledge Pavel Škaloud (Department of Botany, Charles University, Prague, Czechia) for providing the CAUP algal cultures and valuable advice; Norbert Zlámal (Botanical Garden, Comenius University, Bratislava, Slovakia) for identifying and providing *Selaginella uncinata*; Marie Korecká (Institute of Experimental Botany, Czech Academy of Sciences, Prague, Czechia) for assistance with experimental work. This work was supported by Czech Science Foundation project no. 20-13587S, and Charles University Grant Agency (GAUK), project no. 393422. D.V.D.S. acknowledges the Research Foundation Flanders (FWO; G082421N) and Ghent University (BOF-BAS) for financial support.

## Author contributions
V.S. and R.S. cultivated the biological material, analyzed data, and wrote the text. K.K. and S.H. cultivated and prepared samples of certain analyzed organisms. R.V. and P.D. performed mass spectrometry. T.D. and A.P. measured ethylene production. V.S. and S.V. performed the statistical analysis. V.M., D.V.D.S. and J.P. supervised the work and writing.

## Competing interests
The authors declare no competing interests.
