## [Peer Review File · Nature Communications]

Phytohormone profiling in an evolutionary frameworkReviewer #1 (Remarks to the Author):

Review for Schmidt et al 2023 Nat Comm

Phytohormones are key signaling molecules that sense and allow plants morpho-physiological responses to various environmental challenges. The field of phytohormone evolution has rapidly progressed with major findings showing that: a) bryophytes use phytohormone signalling pathways in a relatively similar fashion to tracheophytes (i.e common ancestry); b) land plants are a member of the Streptophyta, and thus evolved from a fresh-water charophycean algae-like ancestor; c) genomic studies show that major charophycean algae lineages (e.g. Zygenematophytes, Chara, Klebsormidium, Mesostigma) have some but not all components of hormone biosynthetic and signalling pathways. Thus, during the origin of plants, a series of evolutionary changes produced novel and unparalleled mechanisms to respond to the terrestrial environment. How this evolved de novo and from re-wiring of pre-existing components in charophycean algae is still a long-standing question.

Some progress has been made in the field, and it has been shown that in the land plant ancestor, ancient signalling pathways were co-opted to respond to new ligands (e.g ABA). But in other cases, novel signaling pathways evolved to respond to pre-existing ligands (e.g. auxin) with yet uncharacterized functions in algae. A systematic sampling of algal phytohormone production is needed to understand the ways algae respond to their environment, but also how those responses resemble or differ from land plants and how they could have been co-opted to allow the plant ancestor to colonize terrestrial environments.

The paper produced by Schmidt and collaborators does an excellent job at answering the question of how many of the multiple types of hormones are produced in algae (with an emphasis on Streptophyta). This is achieved by a systematic analysis using LC/MS with sampling guided by phylogenetics and using appropriate controls to distinguish between hormones present in algal tissue (Fig 1) or exuded into their environment (Fig 2). Major findings include the fact that auxin is consistently produced across Streptophyta despite not having a canonical land plant auxin biosynthetic pathway (via tryptophan aminotransferases and YUCCA enzymes), trans-zeatin is a land plant-specific hormone, cis-zeatins and salicylic acid are robustly detected across Streptophyta, and jasmonates/dn-OPDA, ABA and ethylene show inconsistent or weaker detection in their tested conditions.

General comments:

Although the paper does a decent job at detecting hormones it is still unable to address what is the effect/function of at least some of these compounds in charophycean algae. Some of these functions have been \rightarrow postulated, for example, in the case of dn-OPDA and thermotolerance (Monte et al 2020 Current Bio) but a systematic approach would go hand in hand with the results of this paper. Growth of Klebsormidium in auxin has been previously reported (Ohtaka et al 2017 Plant Phys) suggesting it may influence cell growth and proliferation. I think the study would benefit from a systematic characterisation of auxin-treated algae with regards to its influence on growth. This would allow understanding of the ancestral role of auxin and how it was repurposed as a hormone by land plants during land colonization.

The assays done to obtain exuded compounds in the media, is a fantastic way to assess whether any of these phytohormones/compounds mediate the interaction with competitors sharing resources in aquatic media. It has been postulated that phytohormones had ancestral roles as 'quorum sensing' signals (Vosolsobě et al 2020) and it does seem that presence of bacteria changes IAA levels for certain taxa (e.g Chlorokybus), but I wonder if specific exuded compounds are dependent on the competition between bacteria/algae and algae/algae and if this should be systematically tested in future work. Could the authors elaborate on this subject perhaps in the section discussing detection of exudates (1st Paragraph of Discussion).

This study found that trans-zeatin is a land plant-specific compound and thus, an evolutionary novelty. It should be clearly stated in the abstract and discussed in-depth. For example, does this correlate with the presence/absence of IPT (adenylate isopentenyl transferase) and LOG (LONELY

GUY) genes across Streptophyta? This could complement the discussion CKX enzymes in the discussion.

Colour code in figure 2a is slightly confusing, and use of grey to portray both 'absent in biomass' as well as 'absent in blank' should be changed.

Corrections

Row 193: individual should be individually

Row 241, first sentence of discussion is incomplete, it should be joined with subsequent sentence.

Row 263: Reconsider using the term 'bioindustry' as this is not a conventional academic term. 'metabolic' is a possible alternative.

Row 270: the sentence 'rather than richer' is not making much sense, should be 'rather than... showing a lineage specific signature', please elaborate.

Row 271: 'plant' should be changed to 'streptophyte'. This sentence should also be joined the following sentence as it is part of the same idea. Second sentence needs to be simplified as it is a bit clunky.

Row 273 'Auxin is a prime example' should be followed by colon and not a full stop

Row 294. Last sentence of paragraph leaves me without much hope on how we can test the conservation or partial conservation of a hormonal biosynthetic pathway. What alternatives do we have? Thorough functional characterization of (phylogenetically guided) model organisms in charophycean algae, forward and reverse genetic approaches?

Reviewer #2 (Remarks to the Author):

This manuscript makes some useful contributions to understanding embryophyte phytohormone evolution. Over the last few decades, there have been several reports of phytohormone production in green algae. There were often significant differences between the reported concentrations in these papers, which the authors suspect is due to inconsistent growth media, conditions, and detection methods. To address that inconsistency, the authors measured phytohormone concentrations in a number of plants and green algae, referenced against a blank control. The manuscript also includes a comprehensive literature review of studies reporting phytohormone concentrations in streptophytes, chlorophytes, and non-green lineages.

There are significant flaws in the preparation of the manuscript, much of which would have to be extensively rewritten for clarity. Many sentences contain combinations of words that are confusing and obscure the intended meaning. To name a few examples, (L75) "we convincingly detected," (L93) "typically inconsistently among different conditions and strains," and (L189) "nearly universally less abundant." There are also significant problems with the organization; results and discussion are intermingled (despite being separate sections in the manuscript), and there is never a stand-alone presentation of their data. Instead, the manuscript reports measurements and compares those data to what is found elsewhere in the literature but without a direct citation of the source literature. For example, L79-L80 references literature on the detection of ABA in the protonema of *Physcomitrium patens* but provides no citation on that literature. Taken together, the confusing phrasing, structural defects, and haphazard citations mean that what the authors intended to convey can be very difficult to determine.

This heavy reliance on and intermingling of supplementary data with the core of the manuscript prompted me to look carefully at Table S5, which raised a new set of concerns. The table is composed of a combination of qualitative and quantitative information, and the observations underlying these entries are often obscure. Some cells report a numerical value as the concentration, while others have semi-qualitative terms such as "mid-tens," "units-teens," or "0.X-

low units." These qualitative descriptors are not clearly defined. I selected one paper (10.1093/pcp/pcp057) to understand how these qualitative descriptors work. In the analysis of the rice shoots, there is phytohormone data for three mutants and one wild type (Taichung 65). The data table reports ABA concentrations of "low hundred(s)" and IAA concentrations of "mid tens," values which appear to apply only to the *slr1* mutant (ABA = 113.6 ± 5.7 pmol g⁻¹ FW; IAA = 65.77 ± 11.87 pmol g⁻¹ FW). By contrast, the wild type, Taichung 65, was reported to have an ABA and IAA concentration of 178.2 ± 17.4 pmol g⁻¹ FW and 139.09 ± 21.77 pmol g⁻¹ FW, respectively. The *gid1-3* mutant has an ABA and IAA concentration of 48.8 ± 4.5 pmol g⁻¹ FW and 97.92 ± 14.08 pmol g⁻¹ FW; the *gid2-1* mutant has an ABA and IAA concentration of 41.9 ± 4.9 pmol g⁻¹ FW and 61.16 ± 11.08 pmol g⁻¹ FW. Qualitative values are always problematic, but the appearance here is that the reported values were from one particular mutant, not the wild type, which is clearly problematic. I want to give the authors the benefit of the doubt that most of the values in the table are valid, but it illustrates the problem with using semi-qualitative measures of the type given extensively in the table and casts doubt on the validity of the table overall. Although the manuscript does provide a more consistent set of measurements than was previously available and an extensive review of the literature, it does not do very much to advance understanding of phytohormone evolution. The study does perform a systematic screen, which gives greater resolution to the phylogenetic distribution of phytohormone production, but there is little further interpretation of the data. There is no discussion of the significance of the phylogenetic distribution of phytohormones and their precursors, an omission that is complicated by the fact that, in most cases, the biological role of these compounds is unknown in the subject organisms. If they are not functioning as phytohormones, why are they there? What (if anything) is known? The assertion in the abstract that the signaling molecules predate the mechanisms to sense them is interesting, but there is no discussion in the manuscript of mechanisms by which signaling mechanisms might be assembled or of what the phytohormones might be doing in outgroup taxa where the signaling molecule is present, but sensory mechanisms are absent. Perhaps they are serving as signaling molecules but are being detected by a different mechanism. Would we even know?

I want to stress that I am aware that the authors of this paper have done a lot of work to amass their data, that the overall topic is interesting and important, and that their review of the data available in the literature could be helpful if it were cleaned up. However, the problems with this manuscript run deep, and I cannot recommend it for publication in Nature Communications.

Reviewer #3 (Remarks to the Author):

The manuscript entitled "Phytohormone profiling in an evolutionary framework" by Schmidt et al, described the analyses of plant hormones in streptophyte which is the ancestor of land plants. Plant hormones are indispensable for the life of land plants. The origin of plant hormones are big issue of plant science. as far as this reviewer understand, most of the related studies focused on the genes in the genome of streptophytes homologous to land plant genes involved in plant hormone synthesis and signal transduction. Therefore, it is necessary to analyze the presence of plant hormones in those organisms precisely.

The author examined the plant hormone levels of various streptophyte organisms obtained from fields or cultures by means of reliable detection methods using LC/MS.

This study could offer key information and gave an impact on this research field.

However, totally, I am afraid that the data presented in this manuscript is not conclusive. I have several comments.

First, this study contains only the hormone level data of streptophyte. I do not think the just measuring the plant hormone levels is sufficient to conclude that this or that phytohormone is actively functioning or not. Upon determination of the hormone's levels, they can evaluate the physiological activity at those levels in each organism, I guess. Such the data would be required for judging if the detected phytohormones are physiologically relevant or not.

I am concerned how much each organism were isolated. It is clear that algae (or any organisms)

can have symbiotic relations with bacteria or other small organisms. Those symbiotic organisms might synthesize plant hormones such as IAA. The author mentioned that they established "sterile culture" but I do not think this means "isolated" because some symbiotic bacteria cannot be removed easily, they adhere tightly or live almost inside of the cell. I think the author must check it more carefully. For example, sensitive analyses to check contaminations of other organisms in the samples such as PCR based 5S ribosomal RNA analysis can be done.

The author concluded that IAA, CK, ABA are not functioning as hormone in charophytes like in land plants based on their different profile from land plants, and their similarity to those of animals. However, they did not show how different or how similar and where the borders are. They also mentioned that higher doses of IAA and ABA were required for producing phenotypes referring other studies. But it is not clear that these levels are very different from those they detected in this study. In addition, they took the information that canonical receptors for IAA and ABA have not been found in green algae. Those criteria the author used to conclude are quite unclear. The author should clearly show the requirements (plant hormone level, their profiles, gene organization, physiological activity, etc) for their conclusion of "functional plant hormones". I am not sure the hormone receptors or hormone signaling machinery should be conserved well in ancestor organisms. If this argument stands, the author must do the comparative analysis between the gene organization and the plant hormone detection systematically. Such data would be useful for further discussion.

Anyway, the author concluded that most of plant hormones are not functioning as hormones in streptophyte or charyophye. Then they should have an answer or opinion when they established gathering information of their hormones measuring data and available genome information data, they can propose.

Finally, the structure of the manuscript should be improved. The Introduction is less informative. Many related important papers are not referred. There has been already a lot of discussions on phytohormones in charohytes. On the other hands, the Discussion of this manuscript is too long and including the background information related to phytohormones. I think the author should transfer some of the information from Discussion to Introduction. First two page of the discussion sounds introduction story. In addition, it should be noted in the introduction that plant hormone-like compounds also found in various organisms. I think that the most controversial issue on the streptophyte has been whether the genes involved in hormone synthesis or hormone signal transduction are present or not in their genomes (Hori et al. *Nat Commun*, 2014; Ju et al., *Nat Plant*, 2014; Turnaev et al., *TPS*, 2015; Wang et al., *Plant Physiol.*, 2015; Romani FPS, 2017; Marin-Arevalillo et al, *PLOS Gen*, 2019; Blazquez et al, *ARPB*, 2020, so on). The author should explain the history and the present situation or questions about plant hormone-related studies in streptophyte. Explaining it would increases the importance of this study.

RESPONSES TO THE REVIEWERS

We are very grateful for comments from all three reviewers who raised important comments limiting the strength of this study. Before we address all comments, we mention here 4 main points that have been addressed:

1) The manuscript has been significantly rewritten.

In our initial attempt to address a broad topic in a brief form, the manuscript ended up being imbalanced between its individual sections and difficult to read. This has been mentioned in various ways by all reviewers. Reviewer #2 claimed that “much of the manuscript would have to be extensively rewritten for clarity”. We fully agree with this view and hence present a new, extensively rewritten version. Specifically, the Abstract was updated, the Introduction expanded, including information previously stated in the Discussion, parts from Results were moved into the Discussion, and the Discussion was restructured. The new first two paragraphs relate to the information previously included in Results.

2) We performed a test of the axenicity of used cultures.

Reviewer #3 raised concerns regarding the axenicity of cultures. We consulted this with the institutions maintaining the culture collections of algae where we purchased our strains from (UTEX-USA, NIES-Japan, SAG-Germany). Per their collective recommendations, we performed and included in the revised manuscript a culture contamination test using four types of media combined with culture conditions promoting the growth of microorganisms but not the growth of the green algae. We applied this test to both green algae and land plants included in this study. This is now included in Figure S5, which also includes microscopic images of algae and land plants taken immediately prior to sampling for phytohormone analysis.

3) Table S5 has been reworked.

Reviewer #2 justifiably pointed out that Table S5 (the Excel table listing phytohormone concentrations detected in the literature) was significantly flawed. Although we did not commit factual mistakes, we fully agree that the data presentation in the table left much to be desired. We reworked Table S5. In particular, we re-checked all data against the original studies, clearly specified the source of the listed data in the studies (Figure, analyzed material), noted the nature of the original data (units, fresh or dry weight), and corrected the ambiguous entries. When an approximation of numbers had to be made (from dry to fresh weight), we specified exactly how the data was approximated.

4) Phytohormone treatments have been discussed.

Reviewers #1 and #3 raised the possibility of treatment experiments of green algae by externally applied phytohormones. We fully agree this is one of the major lines of evidence in studying the evolution of phytohormone responses, along with phytohormone profiling and genomics. Therefore, we stress this view in the Discussion. However, treatments with all studied phytohormones would constitute a considerable amount of work beyond the scope of this manuscript. Coincidentally with the suggestion of Reviewer #1, we are currently undergoing such work with regard to auxin in several strains of charophyte algae, which is going to be published separately from this work. Among other things, we observed strain-specific reactions to auxin treatments, narrow concentration differences between no significant reaction and a severe one, and a decisive influence of culture conditions. For these reasons, we believe that treatment experiments ought to be performed separately from this study to be informative. We hope that the currently presented manuscript can help to guide such efforts.

Reviewer 1:

Phytohormones are key signaling molecules that sense and allow plants morpho-physiological responses to various environmental challenges. The field of phytohormone evolution has rapidly progressed with major findings showing that: a) bryophytes use phytohormone signalling pathways in

a relatively similar fashion to tracheophytes (i.e common ancestry); b) land plants are a member of the Streptophyta, and thus evolved from a fresh-water charophycean algae-like ancestor; c) genomic studies show that major charophycean algae lineages (e.g. Zygenematophytes, Chara, Klebsormidium, Mesostigma) have some but not all components of hormone biosynthetic and signalling pathways. Thus, during the origin of plants, a series of evolutionary changes produced novel and unparalleled mechanisms to respond to the terrestrial environment. How this evolved de novo and from re-wiring of pre-existing components in charophycean algae is still a long-standing question.

Some progress has been made in the field, and it has been shown that in the land plant ancestor, ancient signalling pathways were co-opted to respond to new ligands (e.g ABA). But in other cases, novel signaling pathways evolved to respond to pre-existing ligands (e.g. auxin) with yet uncharacterized functions in algae. Algae respond to their environment, but also how those responses resemble or differ from land plants and how they could have been co-opted to allow the plant ancestor to colonize terrestrial environments.

The paper produced by Schmidt and collaborators does an excellent job at answering the question of how many of the multiple types of hormones are produced in algae (with an emphasis on Streptophyta). This is achieved by a systematic analysis using LC/MS with sampling guided by phylogenetics and using appropriate controls to distinguish between hormones present in algal tissue (Fig 1) or exuded into their environment (Fig 2). Major findings include the fact that auxin is consistently produced across Streptophyta despite not having a canonical land plant auxin biosynthetic pathway (via tryptophan aminotransferases and YUCCA enzymes), trans-zeatin is a land plant-specific hormone, cis-zeatins and salicylic acid are robustly detected across Streptophyta, and jasmonates/dn-OPDA, ABA and ethylene show inconsistent or weaker detection in their tested conditions.

We thank the reviewer 1 for the generous review and the many constructive comments.

General comments:

Although the paper does a decent job at detecting hormones it is still unable to address what is the effect/function of at least some of these compounds in charophycean algae. Some of these functions have been postulated, for example, in the case of dn-OPDA and thermotolerance (Monte et al 2020 Current Bio) but a systematic approach would go hand in hand with the results of this paper. Growth of Klebsormidium in auxin has been previously reported (Ohtaka et al 2017 Plant Phys) suggesting it may influence cell growth and proliferation. I think the study would benefit from a systematic characterisation of auxin-treated algae with regards to its influence on growth. This would allow understanding of the ancestral role of auxin and how it was repurposed as a hormone by land plants during land colonization.

We fully agree that treatments constitute a major line of evidence, complementary to our profiling. We stress this view in the discussion. However, in our opinion this would constitute a considerable amount of work beyond the scope of this manuscript. We are currently performing such work with regard to auxin in several strains of charophyte algae, which will be published separately from this work. Our confidential observations so far have included strain-specific reactions to auxin treatments, narrow concentration differences between no significant reaction and a severe one, and a decisive influence of culture conditions. For these reasons, we believe that treatment experiments ought to be performed separately from this study, in order to be addressed properly. We are convinced that the currently presented manuscript can help guide similar efforts, also on effects of other hormones.

The assays done to obtain exuded compounds in the media, is a fantastic way to assess whether any of these phytohormones/compounds mediate the interaction with competitors sharing resources in aquatic media. It has been postulated that phytohormones had ancestral roles as ‘quorum sensing’ signals (Vosolsobě et al 2020) and it does seem that presence of bacteria changes IAA levels for certain taxa

(e.g Chlorokybus), but I wonder if specific exuded compounds are dependent on the competition between bacteria/algae and algae/algae and if this should be systematically tested in future work. Could the authors elaborate on this subject perhaps in the section discussing detection of exudates (1st Paragraph of Discussion).

We thank the reviewer for this comment and discuss this issue further in paragraph #3 of the discussion in the revised manuscript.

This study found that trans-zeatin is a land plant-specific compound and thus, an evolutionary novelty. It should be clearly stated in the abstract and discussed in-depth. For example, does this correlate with the presence/absence of IPT (adenylate isopentenyl transferase) and LOG (LONELY GUY) genes across Streptophyta? This could complement the discussion CKX enzymes in the discussion.

We thank for this comment. We stressed the unique production of trans-zeatin in land plants in the revised abstract. Moreover, we provide a more in-depth discussion in paragraph #4 of Discussion of the revised manuscript.

Colour code in figure 2a is slightly confusing, and use of grey to portray both ‘absent in biomass’ as well as ‘absent in blank’ should be changed.

The colour code was improved.

Corrections:

Row 193: individual should be individually

Row 241: first sentence of discussion is incomplete, it should be joined with subsequent sentence.

Row 263: Reconsider using the term ‘bioindustry’ as this is not a conventional academic term. ‘metabolic’ is a possible alternative.

Row 270: the sentence ‘rather than richer’ is not making much sense, should be ‘rather than... showing a lineage specific signature’, please elaborate.

Row 271: ‘plant’ should be changed to ‘streptophyte’. This sentence should also be joined the following sentence as it is part of the same idea. Second sentence needs to be simplified as it is a bit clunky.

Row 273: ‘Auxin is a prime example’ should be followed by colon and not a full stop

Row 294: Last sentence of paragraph leaves me without much hope on how we can test the conservation or partial conservation of a hormonal biosynthetic pathway. What alternatives do we have? Thorough functional characterization of (phylogenetically guided) model organisms in charophycean algae, forward and reverse genetic approaches?

All suggested corrections were addressed in the manuscript, as far as they remained in the considerably re-written text.

Reviewer 2:

This manuscript makes some useful contributions to understanding embryophyte phytohormone evolution. Over the last few decades, there have been several reports of phytohormone production in green algae. There were often significant differences between the reported concentrations in these papers, which the authors suspect is due to inconsistent growth media, conditions, and detection methods. To address that inconsistency, the authors measured phytohormone concentrations in a number of plants and green algae, referenced against a blank control. The manuscript also includes a comprehensive literature review of studies reporting phytohormone concentrations in streptophytes, chlorophytes, and non-green lineages.

There are significant flaws in the preparation of the manuscript, much of which would have to be extensively rewritten for clarity.

We thank reviewer 2 for the constructive critique which helped us to make significant improvements to our manuscript, compared to the version that we initially submitted.

The manuscript layout and lack of clarity were rightfully criticized by both reviewers 2 and 3 and we are very happy for their constructive comments. We fully agree with this view and hence present a new, extensively re-written version. Specifically, we have updated the abstract, the introduction was expanded to include some information previously stated in the discussion, parts of the results were moved to the discussion, and the discussion itself was restructured. The new first two paragraphs of the discussion relate to the information previously included in the results. The parts on the origins of phytohormone identity (paragraphs 5 and 6 of the revised discussion) were substantially rewritten.

Many sentences contain combinations of words that are confusing and obscure the intended meaning. To name a few examples, (L75) “we convincingly detected,” (L93) “typically inconsistently among different conditions and strains,” and (L189) “nearly universally less abundant.”

We fully agree and took care to eliminate these ambiguous terms from the revised manuscript.

There are also significant problems with the organization; results and discussion are intermingled (despite being separate sections in the manuscript), and there is never a stand-alone presentation of their data.

As mentioned in the previous response, the results section in the revised text pertains directly to our own results. Much of the points previously discussed in the results were, appropriately as suggested, moved into the revised Discussion (initial three paragraphs).

Instead, the manuscript reports measurements and compares those data to what is found elsewhere in the literature but without a direct citation of the source literature. For example, L79-L80 references literature on the detection of ABA in the protonema of *Physcomitrium patens* but provides no citation on that literature. Taken together, the confusing phrasing, structural defects, and haphazard citations mean that what the authors intended to convey can be very difficult to determine.

Many thanks for this comment, we took care to avoid such missteps in the revised version of the manuscript.

This heavy reliance on and intermingling of supplementary data with the core of the manuscript prompted me to look carefully at Table S5, which raised a new set of concerns. The table is composed of a combination of qualitative and quantitative information, and the observations underlying these entries are often obscure. Some cells report a numerical value as the concentration, while others have semi-qualitative terms such as “mid-tens,” “units-teens,” or “0.X-low units.” These qualitative descriptors are not clearly defined. I selected one paper (10.1093/pcp/pcp057) to understand how these qualitative descriptors work. In the analysis of the rice shoots, there is phytohormone data for three mutants and one wild type (Taichung 65). The data table reports ABA concentrations of “low hundred(s)” and IAA concentrations of “mid tens,” values which appear to apply only to the *slr1* mutant (ABA = 113.6 ± 5.7 pmol g⁻¹ FW; IAA = 65.77 ± 11.87 pmol g⁻¹ FW). By contrast, the wild type, Taichung 65, was reported to have an ABA and IAA concentration of 178.2 ± 17.4 pmol g⁻¹ FW and 139.09 ± 21.77 pmol g⁻¹ FW, respectively. The *gid1-3* mutant has an ABA and IAA concentration of 48.8 ± 4.5 pmol g⁻¹ FW and 97.92 ± 14.08 pmol g⁻¹ FW; the *gid2-1* mutant has an ABA and IAA concentration of 41.9 ± 4.9 pmol g⁻¹ FW and 61.16 ± 11.08 pmol g⁻¹ FW. Qualitative values are always problematic, but the appearance here is that the reported values were from one particular mutant, not the wild type, which is clearly problematic. I want to give the authors the benefit of the doubt that most

of the values in the table are valid, but it illustrates the problem with using semi-qualitative measures of the type given extensively in the table and casts doubt on the validity of the table overall.

We fully agree that the initially submitted version Table S5 (a table of phytohormone concentrations measured in literature) was flawed in many respects. We have reworked Table S5. In particular, we re-checked all data against the original studies, clearly specified the source of the listed data in the studies (Figure, analyzed material), noted the nature of the original data (units, fresh or dry weight) and corrected the ambiguous entries. When an approximation of numbers had to be made (from dry to fresh weight), we specified exactly how the data was approximated.

Concerning the study mentioned (10.1093/pcp/pcp057), we have arrived at a misunderstanding, caused by our mistake in the ambiguity of data presentation. Specifically, we listed data from Figure 3 in the original study, whereas you focused on the data listed in Table S4 of the original study. We failed to properly state the source of the listed data in the original study. But in this specific case, we also made the mistake of labeling the source organ as “shoot” in the Figure 3 of the original study, the data is not present for “shoots” but rather broken down into individual organs of the shoot. In Table S4 of the original study that you focused on, the organ is listed as “whole shoots”, which is unfortunately very similar to our summary term “shoot” that we used when listing the data from the original Figure 3. Hence we did not present wrong numbers, but failed to list the information necessary to properly trace the data back into the initial studies. This was corrected in the revised Table S5, as listed above.

Although the manuscript does provide a more consistent set of measurements than was previously available and an extensive review of the literature, it does not do very much to advance understanding of phytohormone evolution. The study does perform a systematic screen, which gives greater resolution to the phylogenetic distribution of phytohormone production, but there is little further interpretation of the data. There is no discussion of the significance of the phylogenetic distribution of phytohormones and their precursors, an omission that is complicated by the fact that, in most cases, the biological role of these compounds is unknown in the subject organisms. If they are not functioning as phytohormones, why are they there? What (if anything) is known?

We thank the reviewer 2 very much for this comment. Indeed, the information on this topic is sparse. We expanded the discussion on interpreting the presence of “phytohormone” compounds in taxa where phytohormone responses are unknown, perhaps missing. This is now paragraph 6 of the revised Discussion.

The assertion in the abstract that the signaling molecules predate the mechanisms to sense them is interesting, but there is no discussion in the manuscript of mechanisms by which signaling mechanisms might be assembled or of what the phytohormones might be doing in outgroup taxa where the signaling molecule is present, but sensory mechanisms are absent. Perhaps they are serving as signaling molecules but are being detected by a different mechanism. Would we even know?

As you point out, we are limited in our inferences by certain issues, including a lack of exploration of Viridiplantae lineages such as green algae and our bias in viewing plant evolution through the pathways known from angiosperms. We mention this at the end of paragraphs 4 and 5 in the revised discussion. Moreover, we stress that the biosynthesis of compounds recognized as signaling molecules needs not to signify a signaling function of its own (see also previous response). As stated, we would not know whether organisms such as green algae can perceive certain phytohormones by independently-evolved means. However, research in bryophytes has recently begun to uncover the presence of possibly ancient mechanisms of phytohormone action, independent of the signaling known from angiosperms, and it is possible that future research will expand this knowledge into charophyte

algae, once model organisms are properly established. This is now discussed in paragraph 5 of the revised discussion.

I want to stress that I am aware that the authors of this paper have done a lot of work to amass their data, that the overall topic is interesting and important, and that their review of the data available in the literature could be helpful if it were cleaned up. However, the problems with this manuscript run deep, and I cannot recommend it for publication in Nature Communications.

We are very grateful for the comprehensive review of our work and for the clear formulation of concerns. It is now our hope that the revised and re-written version of our manuscript has addressed the concerns raised.

Reviewer 3:

The manuscript entitled “Phytohormone profiling in an evolutionary framework” by Schmidt et al, described the analyses of plant hormones in streptophyte which is the ancestor of land plants. Plant hormones are indispensable for the life of land plants. The origin of plant hormones are big issue of plant science. as far as this reviewer understand, most of the related studies focused on the genes in the genome of streptophytes homologous to land plant genes involved in plant hormone synthesis and signal transduction. Therefore, it is necessary to analyze the presence of plant hormones in those organisms precisely. The author examined the plant hormone levels of various streptophyte organisms obtained from fields or cultures by means of reliable detection methods using LC/MS. This study could offer key information and gave an impact on this research field. However, totally, I am afraid that the data presented in this manuscript is not conclusive. I have several comments.

We thank reviewer 3 for the constructive critique which helped us to make significant improvements to our manuscript, compared to the version that we initially submitted. As part of a major revision requested by the editor, we substantially re-wrote the manuscript to address the concerns raised by specifically by reviewers 2 and 3 and generally to improve the clarity of the text. Please see also the beginning of the response to reviewer 2.

First, this study contains only the hormone level data of streptophyte. I do not think the just measuring the plant hormone levels is sufficient to conclude that this or that phytohormone is actively functioning or not.

We fully agree and thank for this comment. It was not our direct aim to test whether the compounds known as phytohormones of land plants also have a physiological/developmental function in charophyte algae, especially given the large number of phytohormones we screened these organisms for. This would require a multitude of other studies, each preferably limited to one phytohormone class. The establishment of several charophyte model organisms in laboratory conditions is a desirable prerequisite for this effort. Rather, we consider our study as one contribution among the multiple lines of evidence necessary to illuminate the evolution of phytohormone identity in the green lineage.

Upon determination of the hormone’s levels, they can evaluate the physiological activity at those levels in each organism, I guess. Such the data would be required for judging if the detected phytohormones are physiologically relevant or not.

We fully agree that treatments in order to identify the effects constitute a major line of evidence, complementary to our profiling. We stress this view in the discussion. However, in our opinion this would constitute a considerable amount of work beyond the scope of this manuscript. We are currently

performing such work with regard to auxin in several strains of charophyte algae, which will be published separately from this work. Our confidential observations so far have included strain-specific reactions to auxin treatments, narrow concentration differences between no significant reaction and a severe one, and a decisive influence of culture conditions. For these reasons, we believe that treatment experiments ought to be performed separately from this study, in order to be addressed properly. We hope at least that the currently presented manuscript can help guide such similar efforts.

I am concerned how much each organism were isolated. It is clear that algae (or any organisms) can have symbiotic relations with bacteria or other small organisms. Those symbiotic organisms might synthesize plant hormones such as IAA. The author mentioned that they established “sterile culture” but I do not think this means “isolated” because some symbiotic bacteria cannot be removed easily, they adhere tightly or live almost inside of the cell. I think the author must check it more carefully. For example, sensitive analyses to check contaminations of other organisms in the samples such as PCR-based 5S ribosomal RNA analysis can be done.

We are grateful for this comment and would like to clarify the origin of all cultures. We did not isolate algal cultures ourselves but rather obtained these from institutions that maintain culture collections, as mentioned in Methods. We consulted the options of testing the axenicity of our cultures with these institutions (UTEX-USA, NIES-Japan, SAG-Germany). Based on their recommendations, we performed a culture contamination test by culturing our green algae and land plants on four types of media, combined with specific culture conditions, designed to promote the growth of microorganisms (bacteria, fungi) but not the growth of the green algae or plants. This information is now included as Figure S5, which comprises photographs of plates and microscopic images of their contents, in case contamination was detected. We also include in Figure S5 microscopic images of algae as well as photographs of land plants, taken immediately prior to sampling for phytohormone analysis.

The author concluded that IAA, CK, ABA are not functioning as hormone in charophytes like in land plants based on their different profile from land plants, and their similarity to those of animals. However, they did not show how different or how similar and where the borders are.

We thank the reviewer for the comment. In the previous version, we presented the hypothesis on IAA, CK, ABA phytohormone function originating in land plants based on multiple factors. We agree the conditions to reach this conclusion had to be better defined. This is now elaborated in paragraph 5 of the discussion in the revised text. We hope it will be satisfactory.

They also mentioned that higher doses of IAA and ABA were required for producing phenotypes referring other studies. But it is not clear that these levels are very different from those they detected in this study.

Many thanks for this comment. We now improved the clarity of the statement on this issue in the last part of the discussion. We believe that it is difficult to directly compare concentrations from treatment experiments to our data. The treatment studies only state the initial concentration of the applied phytohormone in the culture medium, but never measured the actual intracellular concentrations in cells that developed the phenotypes. This is certainly an issue that should be addressed more properly in future studies.

In addition, they took the information that canonical receptors for IAA and ABA have not been found in green algae. Those criteria the author used to conclude are quite unclear. The author should clearly show the requirements (plant hormone level, their profiles, gene organization, physiological activity, etc) for their conclusion of “functional plant hormones”.

We thank reviewer 3 for this comment. The analysis of the conservation of canonical IAA and ABA receptors has been performed previously, as cited. We fully agree that the requirements for plant hormone identity had to be better defined in our text. In paragraph 5 of the revised discussion, we discuss the definition of “phytohormones” and by what criteria we may currently infer a conserved “phytohormone function”.

I am not sure the hormone receptors or hormone signaling machinery should be conserved well in ancestor organisms. If this argument stands, the author must do the comparative analysis between the gene organization and the plant hormone detection systematically. Such data would be useful for further discussion.

We absolutely agree with this comment, charophyte green algae have undergone a long time of independent evolution. The possibility exists that an ancient, derived, or even independently evolved form of phytohormone response(s) may be present in some extant green algae. We now allude to this possibility in paragraph 5 of the discussion part.

Anyway, the author concluded that most of plant hormones are not functioning as hormones in streptophyte or charophyte. Then they should have an answer or opinion when they established gathering information of their hormones measuring data and available genome information data, they can propose.

We present this opinion only for auxin, cytokinins and abscisic acid. For other phytohormones we believe there is currently not enough published information to raise such hypotheses. This is now elaborated in paragraph 5 of the discussion in the revised text.

Finally, the structure of the manuscript should be improved. The Introduction is less informative. Many related important papers are not referred. There has been already a lot of discussions on phytohormones in charophytes. On the other hands, the Discussion of this manuscript is too long and including the background information related to phytohormones. I think the author should transfer some of the information from Discussion to Introduction. First two page of the discussion sounds introduction story.

We are very grateful for this comment, which allowed us to improve the clarity of our text. The manuscript layout and lack of clarity were rightfully criticized by reviewers 2 and 3. Based on their valuable points, we present a new, extensively rewritten version. Specifically, the abstract was updated and the introduction was expanded, including information previously stated in the discussion. Parts from the results were moved into the discussion, and the discussion itself was restructured. The new first two paragraphs now relate to the information previously included in the results. The parts on the origins of phytohormone identity (paragraphs 5 and 6 of the revised discussion) were substantially rewritten.

In addition, it should be noted in the introduction that plant hormone-like compounds also found in various organisms.

Thanks for this, we added this to the introduction. Additionally, this issue is discussed in paragraph 6 of the discussion of the revised manuscript.

I think that the most controversial issue on the streptophyte has been whether the genes involved in hormone synthesis or hormone signal transduction are present or not in their genomes (Hori et al. Nat Commun, 2014; Ju et al., Nat Plant, 2014; Turnaev et al., TPS, 2015; Wang et al., Plant Physiol., 2015; Romani FPS, 2017; Marin-Arevalillo et al, PLOS Gen, 2019; Blazquez et al, ARPB, 2020, so on). The author should explain the history and the present situation or questions about plant hormone-related studies in streptophyte. Explaining it would increase the importance of this study.

We fully agree that the recent history in charophyte genomics has not been sufficiently explained in the original submission. We have elaborated on this in the revised introduction, although we could not cite all the important studies on this topic due to limitations in space and the total reference limit. Additionally, we interpret our phytohormone profile in Viridiplantae to the available genomic knowledge in the discussion (paragraph 4).

Reviewer #1 (Remarks to the Author):

The revised manuscript by Schmidt and collaborators improves on the writing and data presentation particularly in the introduction and discussion sections of the manuscript. However, I find that the core structure and limitations of the original manuscript were largely preserved. I am disappointed that a request to include physiological responses to auxin was dismissed as it would have addressed whether the production of such compounds have any biological relevance. The authors show that charophycean algae can produce compounds that we categorize as land plant hormones but choose not to further interrogate such finding: Do charophycean algae use these compounds as hormones themselves? Or are they simply by-products of other metabolic pathways? Is there any conserved function in charophytes that would show the gradual assembly of an environmental response robustly established later on in embryophytes? It is worth mentioning that a recent paper published in *Cell* (Kuhn et al, 2024) has uncovered deeply conserved auxin-mediated phosphorylation responses across charophycean algae (*Klebsormidium*, *Penium*), bryophytes (*Physco/Marchantia*) and tracheophytes (*Arabidopsis*). Thus, at least in the case of auxin, there seems to be an ancient response machinery that predates the land plant-specific nuclear signalling and the canonical Trp-IPyA-dependent biosynthetic pathways. The paper presented here by Schmidt and collaborators could still be a significant contribution to the field if they show a broader and shared physiological role of auxin across all the species sampled so far. Is this phosphor-response used to modulate multicellular growth or perhaps inter/intra-specific interactions? When in evolution did auxin become a ligand assessing changes in light or gravity regimes? In summary, a shortfall of the article is perhaps that the paper 'bit more than it could chew', as the detection of each hormone would require a cumbersome physiological validation for all compounds tested across relevant environmental conditions. An enormous task added to all the work already put in place by the authors. I think that the authors face the choice of either reducing their focus and characterising individual hormonal scenarios in depth, or simply reporting their finding as originally intended but in a journal that meets the limitations of the study.

Reviewer #2 (Remarks to the Author):

The newly submitted manuscript is significantly improved and is nearly acceptable for publication. The authors have rewritten the paper, fixed formatting problems, and increased the readability of their supplemental data. However, in many places, the language used undermines the paper's message. I have provided some suggestions and corrections below, but a careful reading of the paper would be wise (I doubt that I have found everything).

Corrections:

L68-70: This sentence is a little hard to understand. Are you saying it is unclear if new charophyte genomes contain biosynthetic pathways for phytohormones?

L85-86: I think you mean that the objective of the study is to study charophyte algae, not chlorophytes as is currently written.

L142-143: *Klebsormidiophyceae* and should not be italicized. Only components of a binomial name (genus/species) should be italicized.

L173: "*Chara ssp.*" means *Chara* subspecies. You probably mean *Chara* spp. (multiple species of *Chara*)

L222: *Klebsormidiophyceae* and should not be italicized.

L244-247: *Mesostigmatophyceae*, *Chlorokybophyceae*, and *Coleochaetophyceae* should not be italicized.

L281: *Streptophyta* should not be italicized.

L284-285: "Considerably mutually inconsistent" is confusing wording. It could be replaced by simply "inconsistent."

L294: Once again, *Streptophyta* should not be italicized.

L303-305: This sentence is rough to read; I would word it as something like "It has previously been suggested that the excretion of auxin in charophytes may facilitate inter-organismal communication, and that this might be its ancestral role."

L323-326: This is a great idea, but this sentence is difficult to read in its current form.

L333: You probably mean *Klebsormidium* spp. instead of *Klebsormidium ssp.*

L345: I found the phrase "full gene complements" confusing; I think it would be clearer to say "all of the genes of the pathway" or something similar.

L347: Is it only incomplete in the chlorophytes or the charophytes as well? This should be made clear.

L349: It seems like you are talking about more than just non-seed plants as they were not the main subject of this study. Consider changing the phrasing to "in lineages outside of seed plants."

L350: This sentence should be reworded for clarity. Consider something like "Understanding plant evolution through the prism of knowledge limited to angiosperms..."

L352-354: I disagree with this definition of hormone. The definition of hormone that I would use would be a molecule that 1) is produced at low concentration, 2) is transported (actively or passively), and 3) induces an effect elsewhere. A pheromone is a hormone that is produced by one organism and induces change in another of the same species. If two different species are involved, then it is a signaling molecule, but not technically a pheromone. If the molecule is not produced by an organism, then it is an environmental response. For example, in cyanobacteria ethylene appears to be an environmental signaling molecule, not a hormone. I would be comfortable using the term "signaling molecule" or "signaling mechanism" to refer to all of these processes.

L354-355: Although I don't agree with the definition of hormone given, the authors make an interesting point here. It seems likely that many hormones were recruited from environmental response mechanisms by the subsequent addition of a biosynthetic mechanism. However, as written, this sentence is confusing and confounds microalgae with phylogenetic lineages. For example, it suggests that chlorophytes are microalgae, but charophytes are not. Additionally, this observation potentially applies to non-microscopic species of algae, so the statement should be reworded to be clearer.

L372-380: This is another example of there being valuable ideas in the manuscript, but the language is sufficiently confusing that it is easy to miss the careful thought behind it. I recommend rewriting these for clarity and conciseness.

L381-382: This sentence should be reworded to "... lack a bona fide phytohormone response." Note my comments above about the definition of various signaling processes.

L382: The word "mutually" is unnecessary here.

L384: You probably mean *Dunaliella* spp.

L425: Charophyceae should not be italicized.

The following are suggestions on the language of your paper. These are not necessary corrections, but they may improve your manuscript:

L46: It is probably better to say, "the last common ancestor of land plants," since technically there is no singular common ancestor of land plants.

L41-45: This sentence is a little wordy. You might want to consider some version of the following sentence "Genomic evidence suggests that the machinery underlying ethylene and CK signaling evolved prior to the emergence of land plants, but it is unknown..."

L51: Like L46, it should be the "last common ancestor."

L50-53: This sentence could be less ambiguous when it comes to mentioning the homologs of "certain enzymes."

L163: Consider removing the word "tested" from this sentence.

L273: I don't love the rhetorical question. This sentence could be eliminated without altering the meaning of the article – and I recommend doing so – but this is really at the discretion of the authors.

L302-303: This sentence could be reworded for clarity.

L319: Consider using the word "contaminants" instead of "microorganisms." Note also that the word "xenic" is available and could be used instead of "contaminated."

L388-390: This sentence is a little clunky and could be reworded.

L392-394: This sentence could be improved for clarity. Consider something like "Additionally, phytohormone bioactivity may be spatiotemporally restricted (e.g., to developing zygospores)."

Reviewer #3 (Remarks to the Author):

Based on the author response, the author agreed with most of my comments. And I found some improvement in the text. I understand it would be difficult to do all the experiments I pointed out. But I am afraid that the presented data sounds just measuring of phytohormones of the publicly

available various charophytes samples. The presented data showed that ABA and JA are not commonly detected. I think these results are interesting. However, regardless all the results are expected or not, if these are novel, additional experiments with different approach or aspects should be done to evaluate them. Since the genome data of several charophytes are available. For example, I think, the author can do RNAseq analysis upon hormone application to such the charophytes.

Reviewer #1 (Remarks to the Author):

The revised manuscript by Schmidt and collaborators improves on the writing and data presentation particularly in the introduction and discussion sections of the manuscript. However, I find that the core structure and limitations of the original manuscript were largely preserved.

RESPONSE: We took care to address all concerns and limitations that were specifically identified in the initial review. If there were any limitations remaining, we would have been happy to address them based on your suggestions.

I am disappointed that a request to include physiological responses to auxin was dismissed as it would have addressed whether the production of such compounds have any biological relevance.

RESPONSE: We respectfully believe that we could not justify showing preferential treatment to auxin among the multiple phytohormones analyzed. Also see Response below.

The authors show that charophycean algae can produce compounds that we categorize as land plant hormones but choose not to further interrogate such finding: Do charophycean algae use these compounds as hormones themselves? Or are they simply by-products of other metabolic pathways? Is there any conserved function in charophytes that would show the gradual assembly of an environmental response robustly established later on in embryophytes?

RESPONSE: Elucidating the biological relevance of phytohormone production was not our stated objective. Importantly, pursuing this would have been out of scope of this work. The evolutionary origins of phytohormone responses in Streptophyta is only approached in discussion (because this is a hot topic in genomic studies), as one of multiple issues. The novelty of our study lies mainly in addressing the gaps and inconsistencies of phytohormone profiling in non-seed plant lineages, particularly in the selection of analyzed lineages, the rich spectrum of analyzed compounds and a thorough discussion of the topic. To avoid a misunderstanding, we further modified the introduction and discussion to define our goals more clearly.

It is worth mentioning that a recent paper published in Cell (Kuhn et al, 2024) has uncovered deeply conserved auxin-mediated phosphorylation responses across charophycean algae (Klebsormidium, Penium), bryophytes (Physco/Marchantia) and tracheophytes (Arabidopsis). Thus, at least in the case of auxin, there seems to be an ancient response machinery that predates the land plant-specific nuclear signalling and the canonical Trp-IPyA-dependent biosynthetic pathways. The paper presented here by Schmidt and collaborators could still be a significant contribution to the field if they show a broader and shared physiological role of auxin across all the species sampled so far. Is this phosphor-response used to modulate multicellular growth or perhaps inter/intra-specific interactions? When in evolution did auxin become a ligand assessing changes in light or gravity regimes?

RESPONSE: Thank you for this comment. We now cite Kuhn et al., 2024 in the Discussion. However, this paper has not shown the native function of this auxin-induced phosphorylation response in algae, nor have the authors provided direct evidence of an auxin response mechanism conserved between green algae and land plants (which would necessitate mutagenesis in algae, currently unavailable). Our manuscript has not been primarily focused on auxin. We do consider the suggested topics exciting, but believe this constitutes work for many years spanning multiple papers. Even auxin treatment of all the charophyte strains in our study would be extremely laborious and out of scope of this manuscript. We further reflect these evolutionary implications in the Discussion.

In summary, a shortfall of the article is perhaps that the paper 'bit more than it could chew', as the detection of each hormone would require a cumbersome physiological validation for all compounds

tested across relevant environmental conditions. An enormous task added to all the work already put in place by the authors. I think that the authors face the choice of either reducing their focus and characterising individual hormonal scenarios in depth, or simply reporting their finding as originally intended but in a journal that meets the limitations of the study.

RESPONSE: A cumbersome physiological validation of the analyzed phytohormones was not the objective of this work. Investigating each hormone separately would necessitate years of work and is out of scope of this study. As we now state in the edited Abstract and Introduction, the main contribution of our work was providing a phylogenetically guided analysis of a broad set of phytohormone-related compounds in cultures well established in laboratory conditions. Many of these strains are currently emerging as model organisms, enabling our data to be used as a reference for future studies. We likewise took great effort to complement our analysis with a critical discussion on the history of phytohormone profiling in non-seed plant lineages. To our knowledge, a deeper insight into this issue has not been available to date.

Reviewer #2 (Remarks to the Author):

The newly submitted manuscript is significantly improved and is nearly acceptable for publication. The authors have rewritten the paper, fixed formatting problems, and increased the readability of their supplemental data. However, in many places, the language used undermines the paper's message. I have provided some suggestions and corrections below, but a careful reading of the paper would be wise (I doubt that I have found everything).

RESPONSE: We thank you for the encouraging assessment. All suggestions below ("corrections" and "not necessary corrections") were addressed by edits in the manuscript. The comment regarding "L352-354" is addressed below by a detailed response.

Corrections:

L68-70: This sentence is a little hard to understand. Are you saying it is unclear if new charophyte genomes contain biosynthetic pathways for phytohormones?

L85-86: I think you mean that the objective of the study is to study charophyte algae, not chlorophytes as is currently written.

L142-143: Klebsormidiophyceae and should not be italicized. Only components of a binomial name (genus/species) should be italicized.

L173: "Chara ssp." means Chara subspecies. You probably mean Chara spp. (multiple species of Chara)

L222: Klebsormidiophyceae and should not be italicized.

L244-247: Mesostigmatophyceae, Chlorokybophyceae, and Coleochaetophyceae should not be italicized.

L281: Streptophyta should not be italicized.

L284-285: "Considerably mutually inconsistent" is confusing wording. It could be replaced by simply "inconsistent."

L294: Once again, Streptophyta should not be italicized.

L303-305: This sentence is rough to read; I would word it as something like "It has previously been suggested that the excretion of auxin in charophytes may facilitate inter-organismal communication, and that this might be its ancestral role.

L323-326: This is a great idea, but this sentence is difficult to read in its current form.

L333: You probably mean Klebsormidium spp. instead of Klebsormidium ssp.

L345: I found the phrase "full gene complements" confusing; I think it would be clearer to say "all of the genes of the pathway" or something similar.

L347: Is it only incomplete in the chlorophytes or the charophytes as well? This should be made clear.

L349: It seems like you are talking about more than just non-seed plants as they were not the main subject of this study. Consider changing the phrasing to "in lineages outside of seed plants."

L350: This sentence should be reworded for clarity. Consider something like "Understanding plant evolution through the prism of knowledge limited to angiosperms..."

L352-354: I disagree with this definition of hormone. The definition of hormone that I would use would be a molecule that 1) is produced at low concentration, 2) is transported (actively or passively), and 3) induces an effect elsewhere. A pheromone is a hormone that is produced by one organism and induces change in another of the same species. If two different species are involved, then it is a signaling molecule, but not technically a pheromone. If the molecule is not produced by an organism, then it is an environmental response. For example, in cyanobacteria ethylene appears to be an environmental signaling molecule, not a hormone. I would be comfortable using the term "signaling molecule" or "signaling mechanism" to refer to all of these processes.

RESPONSE: Our definition of "phytohormone" is a direct quote from Davies, 2010 (doi: 10.1007/978-1-4020-2686-7_1). Other works on this subject are less elaborate, but stress the unique nature of phytohormones that distinguishes them from the "hormones" of other organisms such as animals (Sun et al., 2017 doi.org/10.1016/b978-0-12-811562-6.00013-x; Yamaguchi et al., 2010 ISBN 9780080453828; Agudelo-Morales et al., 2021 doi.org/10.34294/j.jsta.21.10.66). As such, we believe that the original definition of phytohormones in the manuscript is appropriate in the stated context.

L354-355: Although I don't agree with the definition of hormone given, the authors make an interesting point here. It seems likely that many hormones were recruited from environmental response mechanisms by the subsequent addition of a biosynthetic mechanism. However, as written, this sentence is confusing and confounds microalgae with phylogenetic lineages. For example, it suggests that chlorophytes are microalgae, but charophytes are not. Additionally, this observation potentially applies to non-microscopic species of algae, so the statement should be reworded to be clearer.

L372-380: This is another example of there being valuable ideas in the manuscript, but the language is sufficiently confusing that it is easy to miss the careful thought behind it. I recommend rewriting these for clarity and conciseness.

L381-382: This sentence should be reworded to "... lack a bona fide phytohormone response." Note my comments above about the definition of various signaling processes.

L382: The word "mutually" is unnecessary here.

L384: You probably mean *Dunaliella* spp.

L425: Charophyceae should not be italicized.

The following are suggestions on the language of your paper. These are not necessary corrections, but they may improve your manuscript:

L46: It is probably better to say, "the last common ancestor of land plants," since technically there is no singular common ancestor of land plants.

L41-45: This sentence is a little wordy. You might want to consider some version of the following sentence "Genomic evidence suggests that the machinery underlying ethylene and CK signaling evolved prior to the emergence of land plants, but it is unknown..."

L51: Like L46, it should be the "last common ancestor."

L50-53: This sentence could be less ambiguous when it comes to mentioning the homologs of "certain enzymes."

L163: Consider removing the word "tested" from this sentence.

L273: I don't love the rhetorical question. This sentence could be eliminated without altering the meaning of the article – and I recommend doing so – but this is really at the discretion of the authors.

L302-303: This sentence could be reworded for clarity.

L319: Consider using the word "contaminants" instead of "microorganisms." Note also that the word "xenic" is available and could be used instead of "contaminated."

L388-390: This sentence is a little clunky and could be reworded.

L392-394: This sentence could be improved for clarity. Consider something like "Additionally, phytohormone bioactivity may be spatiotemporally restricted (e.g., to developing zygosporoes)."

Reviewer #3 (Remarks to the Author):

Based on the author response, the author agreed with most of my comments. And I found some improvement in the text. I understand it would be difficult to do all the experiments I pointed out. But I am afraid that the presented data sounds just measuring of phytohormones of the publicly available various charophytes samples. The presented data showed that ABA and JA are not commonly detected. I think these results are interesting.

However, regardless all the results are expected or not, if these are novel, additional experiments with different approach or aspects should be done to evaluate them.

Since the genome data of several charophytes are available. For example, I think, the author can do RNAseq analysis upon hormone application to such the charophytes.

RESPONSE: It is true that we have not fully elucidated the origins of phytohormone identity in the green lineage. However, this has not been our objective. Rather, the novelty of our study lies in other aspects. We are the first to have investigated such a wide range of Streptophyta lineages, using a uniform method and for such a broad spectrum of phytohormone compounds. Many of the Streptophyta lineages in this paper have never been similarly investigated before. The existing studies were typically restricted to only a few phytohormone compounds. These limitations have been well recognized by the plant evolutionary community. Finally, in the interpretation of our results we have provided a thorough discussion on the available knowledge of phytohormone profiling in non-seed Viridiplantae, which had been missing.

Based on the additional experiments suggested in the initial review, much stress was laid on the necessity of testing the cultures for contamination; we fully agreed and provided this test as per standard procedures of the Algal Culture Collection institutions. An RNA-seq was not mentioned the initial review, therefore has not been addressed.